# Mesoscale functional organization and connectivity of color, disparity, and naturalistic texture in human second visual area

Hailin Ai[1†], Weiru Lin[2,3†], Chengwen Liu[4,5], Nihong Chen[1,6*], Peng Zhang[2,3,7*]

[1]Department of Psychological and Cognitive Sciences, Tsinghua University, Beijing, China; [2]State Key Laboratory of Brain and Cognitive Science, Institute of Biophysics, Chinese Academy of Sciences, Beijing, China; [3]University of Chinese Academy of Sciences, Changsha, China; [4]Department of Psychology and Cognition and Human Behavior Key Laboratory of Hunan Province, Hunan Normal University, Hunan, China; [5]Center for Mind & Brain Sciences, Hunan Normal University, Changsh, China; [6]THU-IDG/McGovern Institute for Brain Research, Tsinghua University, Beijing, China; [7]Institute of Artificial Intelligence, Hefei Comprehensive National Science Center, Hefei, China

*For correspondence:
rainbowcnh@gmail.com (NC);
zhangpeng@ibp.ac.cn (PZ)

[†]These authors contributed
equally to this work

Competing interest: The authors declare that no competing interests exist.

## eLife Assessment

This study builds on previous findings showing modular organization of primate visual cortical areas by presenting **important** results about the cortical processing of color, disparity, and naturalistic textures in the human visual cortex at the spatial scale of cortical layers and columns using state-of-the-art high-resolution fMRI methods at ultra-high magnetic field strength (7T). **Solid** evidence supports an interesting layer-specific informational connectivity analysis to infer information flow across early visual areas for processing disparity and color signals. While the question of how the modularity of representation relates to cortical hierarchical processing is interesting, the findings that texture does not map onto previously established columnar architecture in V2 is suggestive. The successful application of high-resolution fMRI methods to study the functional organization along cortical columns and layers is relevant to a broad readership interested in general neuroscience.

**Abstract** Although parallel processing has been extensively studied in the low-level geniculostriate pathway and the high-level dorsal and ventral visual streams, less is known at the intermediate-level visual areas. In this study, we employed high-resolution fMRI at 7T to investigate the columnar and laminar organizations for color, disparity, and naturalistic texture in the human secondary visual cortex (V2), and their informational connectivity with lower- and higher-order visual areas. Although fMRI activations in V2 showed reproducible interdigitated color-selective thin and disparity-selective thick 'stripe' columns, we found no clear evidence of columnar organization for naturalistic textures. Cortical depth-dependent analyses revealed the strongest color-selectivity in the superficial layers of V2, along with both feedforward and feedback informational connectivity with V1 and V4. Disparity selectivity was similar across different cortical depths of V2, which showed significant feedforward and feedback connectivity with V1 and V3ab. Interestingly, the selectivity for naturalistic texture was strongest in the deep layers of V2, with significant feedback connectivity from V4. Thus, while local circuitry within cortical columns is crucial for processing color and disparity

information, feedback signals from V4 are involved in generating the selectivity for naturalistic textures in area V2.

## Introduction

A single glance at the world captures a rich amount of visual information. The initial signals hit on the retina are transformed along the visual hierarchy in a way that different aspects of information are processed in parallel streams (*Nassi and Callaway, 2009*). The retino-geniculate-striate pathway of the primate visual system is primarily segregated into magnocellular (M) and parvocellular (P) streams (*Kaplan et al., 1990*; *Lee, 1996*; *Merigan and Maunsell, 1993*), selectively processing different spatiotemporal frequencies of achromatic and chromatic information. In the primary visual cortex (V1), M and P information from the geniculate are transformed into higher-level visual representations, such as motion, disparity, color, and orientation (*Tootell and Nasr, 2017*). Early studies in nonhuman primates established the functional specializations of area V2 (*Burkhalter and Van Essen, 1986*; *Peterhans and von der Heydt, 1989*; *Poggio and Fischer, 1977*; *Zeki, 1973*). The processing of motion/disparity, color, and orientation information has been found to be organized into 'stripe'-shaped interdigitated columns (*DeYoe and Van Essen, 1985*; *Hubel and Livingstone, 1985*; *Hubel and Livingstone, 1987*; *Roe and Ts'o, 1995*; *Ts'o et al., 2001*; *Xiao et al., 2003*), corresponding to the 'thick', 'thin', and 'pale' stripes in cytochrome oxidase (CO) staining studies (*Livingstone and Hubel, 1982*; *Tootell et al., 1983*; referred to as CO-stripes in the present study). The interdigitated columnar organizations for color and motion/disparity processing have also been found in human V2 by high-resolution fMRI at 7T (*Kennedy et al., 2023*; *Nasr et al., 2016*).

More recently, primate V2 was also found to be sensitive to high-order statistical dependencies embedded in naturalistic textures (*Freeman et al., 2013*), a unique type of information critical for surface and material perception. Computationally, these high-order statistical dependencies were composed of correlations across different orientations, spatial scales, and local positions, which can be calculated from the output of orientation filters in V1 (*Freeman et al., 2013*; *Portilla and Simoncelli, 2000*). Although weakly represented in V1 (*Freeman et al., 2013*; *Okazawa et al., 2015*; *Ziemba et al., 2016*), the neural selectivity to naturalistic statistics was found to be increasingly stronger in the downstream areas V3 (*Kohler et al., 2016*) and V4 (*Arcizet et al., 2008*; *Okazawa et al., 2015*; *Okazawa et al., 2017*). Electrophysiological recordings have also identified later and much weaker sensitivity to naturalistic texture in the superficial and deep layers of V1 compared to V2 (*Ziemba et al., 2019*), consistent with an effect of corticocortical feedback modulation. However, it remains unclear whether the neural representations of naturalistic textures arise from local processing within V2 or feedback modulation from higher-order visual areas, such as V4.

While interdigitated stripe-shaped columnar organizations for color and motion/disparity processing have been found in primate V2, it remains unknown whether a columnar organization also exists for texture processing. If local circuits in area V2 are essential for processing high-order naturalistic statistics, specialized cortical columns might develop to enhance computational efficiency (*Mountcastle, 1997*; *Schulte To Brinke et al., 2022*; *Stoop et al., 2013*). A likely candidate for such computational units is the pale stripes, known for their preferential responses to orientation information (*Hubel and Livingstone, 1987*; *Livingstone and Hubel, 1988*; *Lu and Roe, 2008*; *Malach et al., 1994*). Alternatively, if feedback modulations from higher-order visual areas play a prominent role in generating the selectivity in area V2 to high-order statistical dependencies embedded in naturalist textures, a specialized cortical column may not be necessary for such computations. Furthermore, the integration of local elements across various locations, orientations, and spatial scales, which is necessary for processing high-order statistics, might pose a challenge for early visual areas like V2 to develop a specialized computational module.

To better understand the mesoscale functional organizations and neural circuits of information processing in area V2, the present study investigated laminar (or cortical depth-dependent) and columnar response profiles for color, disparity, and naturalistic texture in human V2 using 7T fMRI at 1 mm isotropic resolution. Cortical depth-dependent fMRI (also called laminar fMRI) allows us to noninvasively measure feedforward, local, and feedback activity in human cerebral cortex (*Huber et al., 2017*; *Liu et al., 2021*; *Norris and Polimeni, 2019*; *Olman et al., 2012*; *Polimeni and Uludağ, 2018*). By combining laminar fMRI with informational connectivity methods (*Jia et al., 2020*), we

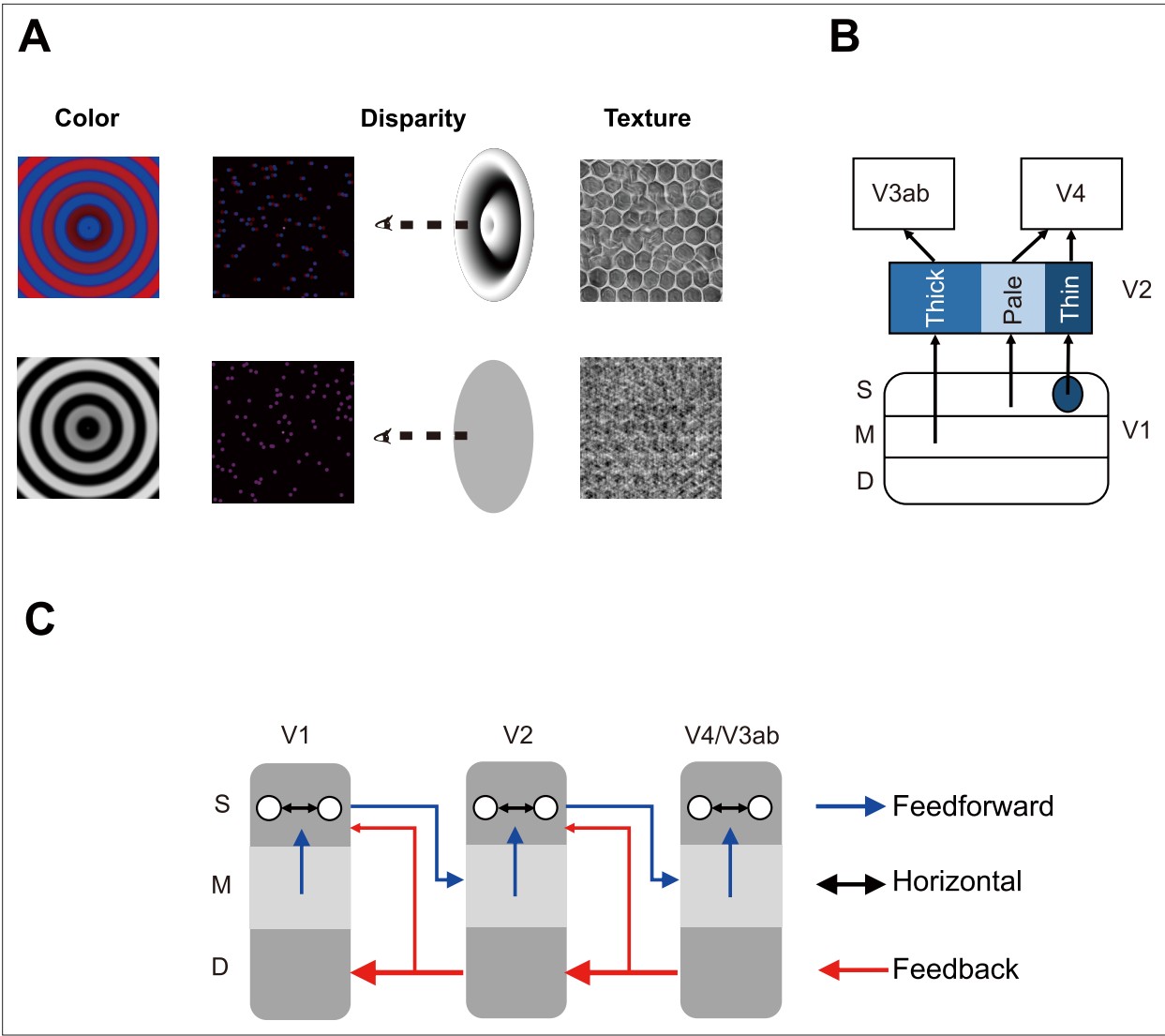

**Figure 1.** Visual stimuli and model of layer-specific neural circuitry. (**A**) Visual stimuli for the fMRI experiments. Left: chromatic and achromatic gratings for the color experiment; Middle: disparity-defined grating and zero-disparity disc from random dots for the disparity experiment; Right: naturalistic texture and spectrally matched noise for the texture experiment. (**B**) Parallel information processing pathways in the early visual areas. (**C**) Layer-specific neural circuitry of feedforward, feedback, and horizontal connections in the early visual areas. S: superficial layers; M: middle layers. D: deep layers.

The online version of this article includes the following figure supplement(s) for figure 1:

**Figure supplement 1.** Results of isoluminance adjustment.

further investigated the feedforward and feedback connectivity between V2 and the lower- and higher-order visual areas. Our results revealed interdigitated stripe-shaped columnar organizations in V2 for color and disparity processing, which involved both feedforward and feedback connectivity with other visual areas in the hierarchy. In contrast, feedback modulations from V4 played a prominent role in processing naturalistic statistics in area V2, which showed no clear evidence of modular organization in CO-stripes.

## Results

*Figure 1A* shows the stimuli for the color, disparity, and texture experiments. Color-selective activation was defined as the contrast of fMRI responses between chromatic (Chr) and achromatic (Ach) gratings (first column). Disparity-selective activation was the response difference between disparity-defined sinusoidal gratings (3D) by random dot stereograms (RDSs) and their zero-disparity (2D)

counterparts (second column). Texture-selective activation was the response difference between naturalistic textures (T) and spectrally matched noise (N) (third column).

*Figure 1B* illustrates a simple model for the building blocks of parallel processing streams in area V2 and their connections with lower (V1) and higher-order visual areas (V3ab and V4). Previous studies in anesthetized macaques found neural selectivity to binocular disparity in the layer 4B of V1, V2 thick stripes, and V3ab in the dorsal stream (*Hubel and Livingstone, 1987*; *Livingstone and Hubel, 1987*; *Tootell et al., 1983*; *Ts'o et al., 2001*; *Tsao et al., 2003*), color selectivity in the color blobs in the V1 superficial layers, V2 thin stripes, and V4 in the ventral stream (*Hubel and Livingstone, 1987*; *Livingstone and Hubel, 1987*; *Livingstone and Hubel, 1988*; *Lu and Roe, 2008*; *Zeki, 1973*), and strong orientation selectivity in layer 2/3 of V1, V2 pale stripes, and V4 (*Hubel and Livingstone, 1987*; *Roe and Ts'o, 1995*; *Tanigawa et al., 2010*; *Ts'o et al., 2001*). Only major connections are shown here. There are also other connections, such as V1 interblobs projecting to thick stripes (*Federer et al., 2021*; *Hu and Roe, 2022*; *Sincich and Horton, 2005*). Given a strong dependency on the output of orientation filters (*Portilla and Simoncelli, 2000*; *Simoncelli and Olshausen, 2001*), naturalistic textures might be selectively processed by orientation-selective neurons in the pale stripes of V2.

*Figure 1C* illustrates a simplified model for the layer-specific neural circuitry in the early visual cortex (*Felleman and Van Essen, 1991*; *Nassi and Callaway, 2009*). In addition to feedforward and local horizontal connections, feedback modulations may also play important roles in processing visual information, especially in conscious visual perception (*Ge et al., 2020*). Here, we focused on major connections in the early visual areas. There are also other connections such as feedforward connections from V1 to V4 (*Ungerleider et al., 2008*), and feedback connections from the superficial layers to lower-order visual areas (*Briggs, 2020*; *Rockland and Pandya, 1979*). Using cortical depth-dependent fMRI, we aim to investigate the feedforward, feedback, and local processing of color, disparity, and texture information in the human visual system.

## Functional organizations on the cortical surface of V2

As shown in *Figure 2A* in a representative subject, color-selective (Chr – Ach, first column, red arrows) and disparity-selective activations (3D – 2D, second column, blue arrows) show stripe-shaped organizations in area V2. These interdigitated stripes can be more clearly seen on the differential map between color and disparity activations [(Chr – Ach) – (3D – 2D), third column]. However, the texture-selective activation map does not exhibit a clear columnar organization (T – N, fourth column). Stronger texture-selective activations can be found from the more anterior part of V2, corresponding to the peripheral visual field. Similar functional organizations can be found from other subjects (*Figure 2— figure supplement 1*).

In 5 out of 10 subjects, both color and disparity experiments were conducted in two sessions. To evaluate the test–retest reliability of the interdigitated columnar organizations, we calculated the inter-session pattern correlations for the color- and disparity-selective functional maps. For the representative subject (*Figure 2B*), the correlation coefficients for both color- ($r = 0.66$) and disparity-selective activation maps ($r = 0.53$) were highly significant (both p<0.001, family-wise-error [FWE] corrected; activation patterns from the two sessions were shown in *Figure 2—figure supplement 2*). The functional maps from the other four subjects also demonstrate highly significant pattern correlations between sessions (*Figure 2—figure supplement 3*). These findings demonstrate that the interdigitated columnar organizations for color and disparity processing are highly reproducible.

To further demonstrate whether there is a difference in texture selectivity within different functional modules of the parallel processing streams in area V2, we performed an ROI analysis with the thick, thin, and pale stripes. The ROIs for disparity-selective thick and color-selective thin stripes were defined by the differential map between color- and disparity-selective activations (color-disparity, the third column in *Figure 2A*), while the ROIs for the pale stripes were defined as vertices in-between adjacent thin- and thick-stripe ROIs (see 'Materials and methods' for details about ROI definition and *Figure 2—figure supplement 4* for the ROIs in a representative subject). From the columnar response profile (*Figure 2D*), thin- and thick-stripe ROIs show the strongest selectivity to color and disparity information, respectively. This is as expected and validated our ROI definition approach. We further conducted repeated-measures ANOVA and Bayesian ANOVA to examine whether there is difference in texture-selectivity index across three different stripes. The results were statistically nonsignificant ($F(2,9) = 1.88$, p=0.18, $BF_{10} = 0.65$). A nonparametric bootstrap method also revealed no significant

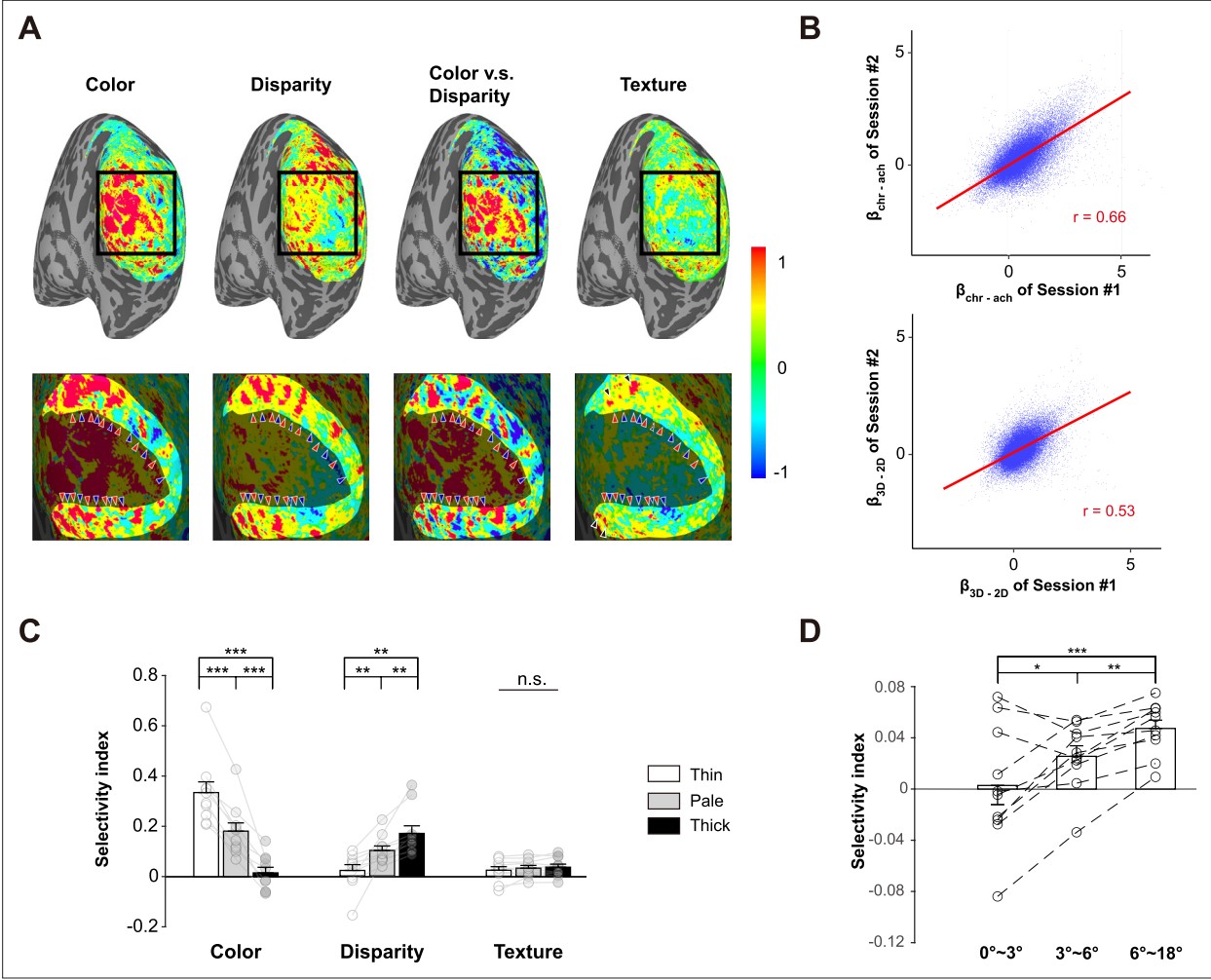

**Figure 2.** Response selectivity of color, disparity, and naturalistic texture in V2. (**A**) Activation maps in a representative subject (S09). The scale bar denotes percent signal change of BOLD response. From left to right: Chr – Ach (color), 3D – 2D (disparity), color – disparity, T – N (texture). The bottom panels show enlarged activations in the black square. The highlighted region in the bottom panels represents area V2. Color-selective and disparity-selective stripe-shaped activations arranged perpendicular to the V1-V2 border. Red arrowheads denote the location of color-selective (thin) stripes and blue arrowheads denote the location of disparity-selective (thick) stripes. Black arrowheads in the fourth column highlight the texture-selective activations in anterior V2 (corresponding to peripheral visual field). The regions of interest (ROIs) for pale stripes were defined as vertices in-between adjacent thin and thick stripes (see 'Materials and methods' for details). (**B**) Inter-session correlations for the color- and disparity-selective functional maps in S09. Each blue dot represents one vertex on V2 surface. (**C**) Selectivity indices for color, disparity and naturalistic texture in different types of columns. Error bar indicates 1 SEM across subjects. **$p<0.01$, ***$p<0.001$. n.s.: none significance. Circles represent data from individual participants. (**D**) Texture selectivity at different eccentricities. Error bars represent 1 SEM across ten participants. *$p<0.05$, **$p<0.01$, ***$p<0.001$.

The online version of this article includes the following source data and figure supplement(s) for figure 2:

**Figure supplement 1.** The functional maps in V2 for all 10 subjects.

**Figure supplement 2.** Test-retest activation patterns for color and disparity experiments in a subject (S09).

**Figure supplement 3.** Inter-session correlations for the color- and disparity-selective activation maps in four subjects who scanned both color and disparity experiments in 2 days.

**Figure supplement 4.** The manually defined ROIs for disparity-selective thick, color-selective thin stripes, and the pale stripes in-between in a representative subject (S09).

**Figure supplement 5.** The bootstrapped distributions of stimulus-selectivity indices in different types of column ROIs.

**Figure supplement 6.** Feature selectivity in CO-stripes under different ROI-definition thresholds.

**Figure supplement 7.** The response difference between texture and noise in the 3T fMRI experiment.

**Figure supplement 8.** Maps showing selectivity for color, disparity and naturalistic texture in V4 and V3ab in a subject (S01).

**Figure supplement 9.** Null distributions of pattern correlation coefficients from Monte Carlo simulation.

*Figure 2 continued on next page*

*Figure 2 continued*

**Figure supplement 1—source data 1.** The number of stripes in manually defined ROIs.

difference between the responses to naturalistic texture and spectrally matched noise (see boot-strapped distributions in *Figure 2—figure supplement 5*). Further analysis shows that the results were independent of the threshold used to define the stripe ROIs (*Figure 2—figure supplement 6*).

Although none of the CO-stripes demonstrated a preference for naturalistic textures, the anterior part of V2 showed some texture-selective activations (*Figure 2A*, *Figure 2—figure supplement 1*). To further demonstrate this observation, we divided V2 into three parts based on eccentricity: 0–3° (central), 3–6° (parafoveal), and 6–18° (peripheral). Texture selectivity increased significantly with eccentricity (*Figure 2D*, F(2,9) = 17.74, p<0.001, $BF_{10}$ = 345.83; peripheral vs. parafoveal: $t(9)$ = 4.12, p<0.01; parafoveal vs. central: $t(9)$ = 3.20, p<0.05). These findings suggest that texture selectivity in V2 is stronger in more peripheral visual fields. However, it is possible that the central visual field might prefer texture patterns with a higher spatial frequency than our stimuli.

Texture-selective activations in our results are weaker compared to those in the previous study (*Freeman et al., 2013*). There also appears to be residual texture patterns in some of the noise stimuli (*Figure 1A*). Thus, to demonstrate the efficacy and specificity of our stimuli, we performed a 3T fMRI experiment using the same stimuli as in Freeman et al.'s study. Different texture families and noise stimuli were presented in separate blocks. *Figure 2—figure supplement 7* shows the 3T results for stimuli that were also used in the 7T experiment. All texture families showed significantly stronger activation in V2 compared to the corresponding noise patterns, even for those that 'appeared' to have residual texture information (e.g., the third texture family). These results demonstrate that our stimuli are effective in activating texture-selective neural populations in area V2. The 3T data also showed a notable increase in texture-selective activations compared to the 7T experiment, likely due to the increased stimulus presentation speed.

## Cortical depth-dependent response selectivity

A response selectivity index was calculated for each stimulus contrast (see 'Materials and methods' for details). The original BOLD responses are also provided (see *Figure 3—figure supplement 1*). Within each ROI, repeated-measures ANOVA was conducted on each type of selectivity index with cortical depth (deep/middle/superficial) as the within-subject factor, followed by post hoc *t*-tests between different depths. Color selectivity was significantly stronger in the superficial cortical depth compared to the middle and deep cortical depths in both V1 (F(2,9) = 15.08, p<0.001, $BF_{10}$ = 133.72; S vs. M: $t(9)$ = 4.70, p<0.01; S vs. D: $t(9)$ = 4.12, p<0.01) and V2 (F(2,9) = 12.93, p<0.001, $BF_{10}$ = 64.83; S vs. M: $t(9)$ = 3.77, p<0.01; S vs. D: $t(9)$ = 3.85, p<0.01). No significant difference was observed across different depths in other visual areas (*Figure 3A*). According to the hierarchical model, the strongest color selectivity in the superficial cortical depth is consistent with the fact that color blobs locate in the superficial layers of V1 (*Figure 1B*, *Felleman and Van Essen, 1991*; *Hubel and Livingstone, 1987*; *Nassi and Callaway, 2009*). The strongest color selectivity in superficial V2 suggests that both local and feedforward connections are involved in processing color information (*Figure 1C*).

Disparity selectivity was significantly higher in the superficial cortical depth compared to the middle and deep cortical depths in V3ab (*Figure 3B*) (F(2,9) = 9.06, p<0.01, $BF_{10}$ = 16.81; S vs. M: $t(9)$ = 3.11, p<0.05; S vs. D: $t(9)$ = 3.75, p<0.01). No significant difference was found across cortical depths in other ROIs (all F(2,9) < 2.85, p>0.08). The absence of laminar difference in disparity selectivity may suggest that both feedforward, feedback, and local mechanisms are involved in processing disparity information in area V2.

Response selectivity to naturalistic texture was strongest in the deep cortical depth in both V1 (F(2,9) = 12.91, p<0.001, $BF_{10}$ = 63.57; D vs. M: $t(9)$ = 2.28, p<0.05; D vs. S: $t(9)$ = 4.76, p<0.01) and V2 (F(2,9) = 8.6, $BF_{10}$ = 14.08, p<0.01; D vs. M: $t(9)$ = 2.49, p<0.05; D vs. S: $t(9)$ = 4.29, p<0.01). Texture selectivity showed no significant difference across cortical depths in the higher-level visual areas V4 (F(2,9) = 1.64, p=0.22) or V3ab (F(2,9) = 3.53, p=0.051), which were significantly larger than those in V1 and V2 (all paired comparisons between ROIs, p<0.001, $BF_{10}$ > 1106.79). Thus, the depth-dependent effect on feature selectivity does not seem to depend on original BOLD response difference. V1 responses to naturalistic textures were also significantly weaker compared to spectrally matched noise, in line with the top-down feedback hypothesis of predictive coding (*Friston,*

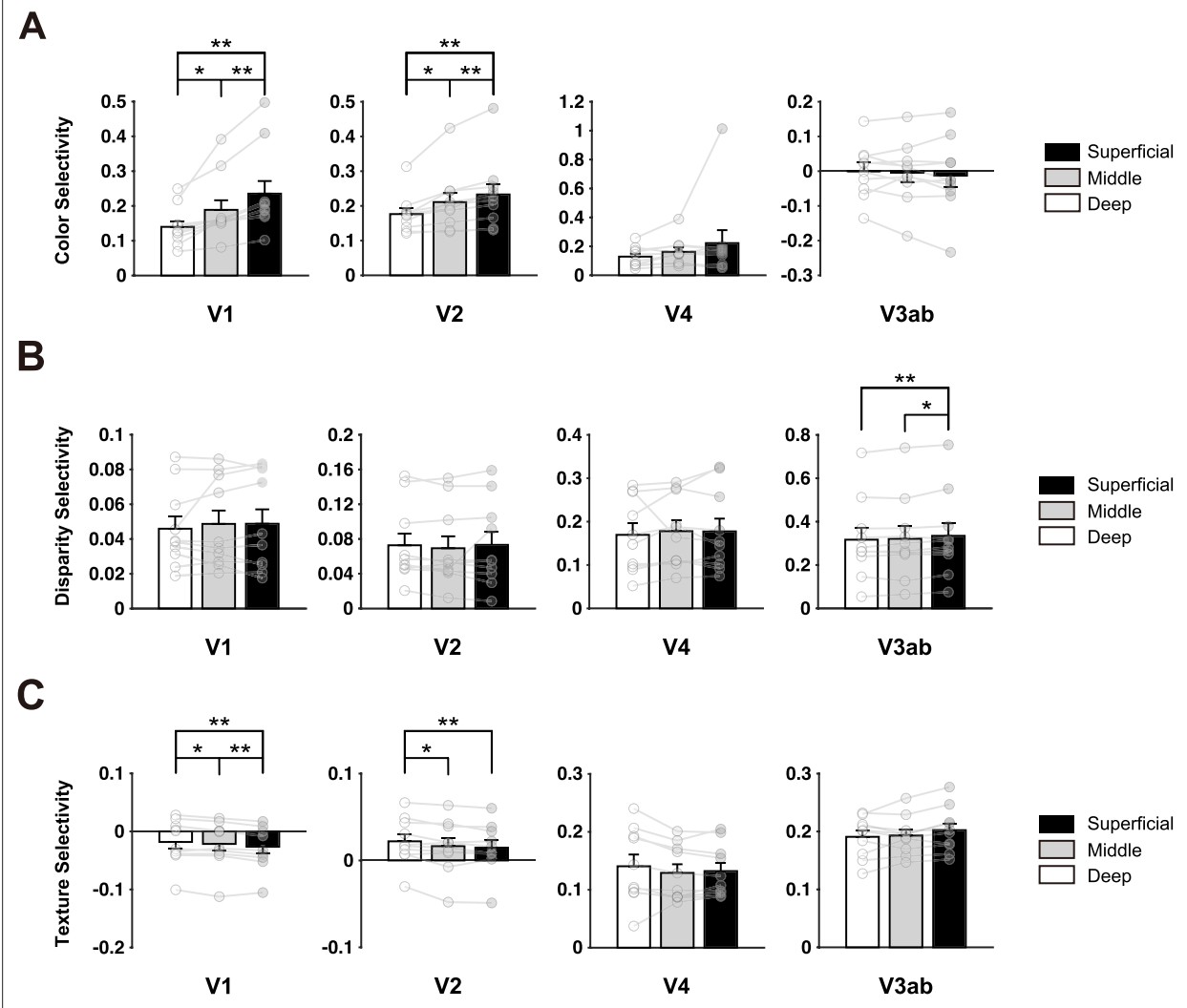

**Figure 3.** Layer-specific response selectivity for color (**A**), disparity (**B**), and naturalistic texture (**C**). Error bars indicate 1 SEM across ten subjects. *p<0.05, **p<0.01. See *Figure 3—figure supplement 1* for the original BOLD response across cortical depth.

The online version of this article includes the following figure supplement(s) for figure 3:

**Figure supplement 1.** Raw BOLD signal changes for calculating layer-specific selectivity indices in color (**A**), disparity (**B**), and texture experiment (**C**).

**Figure supplement 2.** Illustrations of depth map (**A**) and pial vein removal (**B**).

*2005*; *Murray et al., 2002*; *Rao and Ballard, 1999*). The strongest selectivity in the deep layers of V2 suggests that feedback modulations from higher-level visual areas play a crucial role in processing naturalistic statistical information in this region.

## Cortical depth-dependent informational connectivity

To further investigate the information flow in the visual hierarchy, we conducted layer-specific informational connectivity analysis among V1, V2, V3ab, and V4 (*Aly and Turk-Browne, 2016*; *Coutanche and Thompson-Schill, 2014*; *Haxby et al., 2001*; *Huffman and Stark, 2017*; *Jia et al., 2020*; *Koster et al., 2018*). For each pair of stimuli, an support vector machine (SVM) classifier was trained to decode the stimulus type (e.g., chromatic or achromatic gratings). Block-by-block multivariate distances to the decision boundary were used to calculate the co-variability of stimulus representations between two brain regions. Feedforward connectivity was defined as the connection between the superficial layer of a lower-level area and the middle layer of a higher-level area, whereas feedback connectivity was defined as the connection between the deep layers of two brain regions.

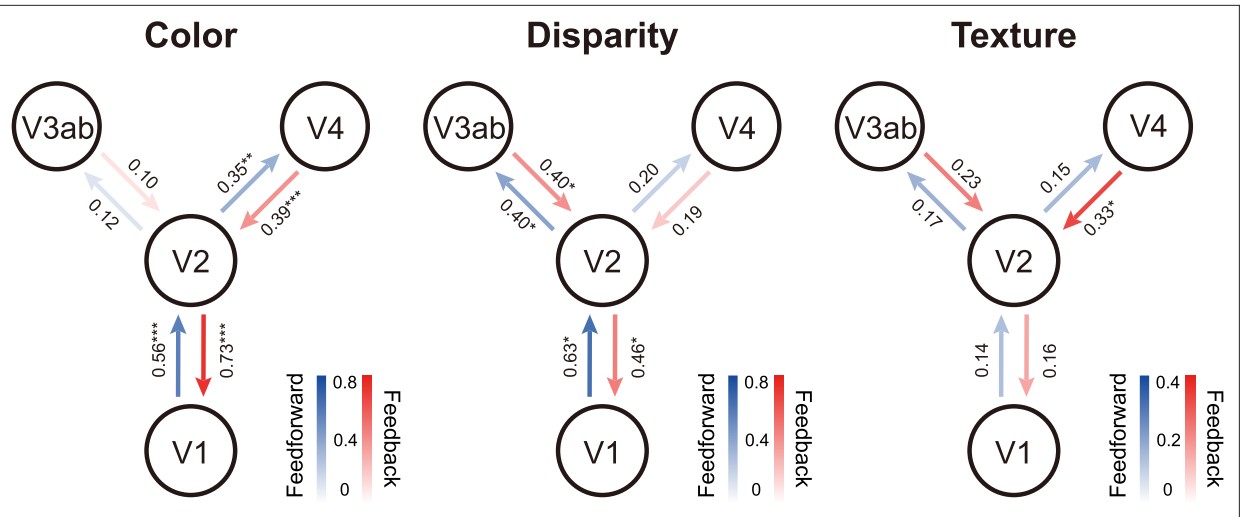

**Figure 4.** Layer-specific feedforward and feedback informational connectivity of color, disparity, and naturalistic texture. Numbers denote the mean values of connection (Pearson's *r*) across all subjects. *p<0.05, **p<0.01, ***p<0.001 after false discovery rate correction.

As shown in *Figure 4*, for color-selective processing, significant feedforward (*t*(9) = 5.64, p<0.001) and feedback connections (*t*(9) = 10.39, p<0.001) were found between V1 and V2, and between V2 and V4 (both *t*(9) > 4.96, p<0.01). For disparity-selective processing, significant feedforward (*t*(9) = 3.77, p<0.05) and feedback connections (*t*(9) = 3.06, p<0.05) were found between V1 and V2, and also between V2 and V3ab (both *t*(9) > 2.90, p<0.05). In contrast, for naturalistic texture-selective processing, a significant feedback connection was found from V4 to V2 (*t*(9) = 4.28, p<0.05). No significant correlation was found for other connections (all *t*(9) < 1.7, p>0.25).

## Discussion

Utilizing 7T BOLD fMRI at 1 mm isotropic resolution, we investigated laminar and columnar response profiles for color, disparity, and naturalistic textures in human V2 by presenting three stimulus contrasts. Color- and disparity-selective activations revealed clear stripe-shaped columnar organizations in area V2, oriented perpendicular to the V1-V2 border (*Figure 2A*). These columnar patterns were reproducible between different scanning sessions (*Figure 2*, *Figure 2—figure supplement 3*) and are consistent with previous findings from intrinsic optical imaging studies in monkeys (*Lu and Roe, 2008*; *Ts'o et al., 2001*) and 7T fMRI studies in humans (*Kennedy et al., 2023*; *Nasr et al., 2016*). However, we found no clear evidence for modular organization for naturalistic textures. CO-stripes in V2 exhibit similar texture selectivity. Cortical depth-dependent analysis revealed that compared to color and disparity information, the processing of naturalistic statistics in V2 is more dependent on feedback modulation from V4.

The laminar profiles of response selectivity revealed both inter- and intra-areal hierarchical processing of visual information. Color selectivity was strongest in the superficial depth of V1 and V2. This result is consistent with the findings that color blobs are primarily located in the superficial layers of primate V1 (*Livingstone and Hubel, 1982*), and that the local processing of color information is most prominent in the superficial layers of the early visual cortex (*Hubel and Livingstone, 1987*; *Lu and Roe, 2008*). For disparity processing, no significant difference was identified across cortical depths in the early visual cortex. This result suggests that feedforward, feedback, and local mechanisms all contribute to generating disparity-defined 3D perception as the middle layer is recognized as the primary termination of feedforward inputs, and the superficial and deep layers are considered as the output layers to higher-order areas and the primary recipients of feedback projections, respectively (*Callaway, 2004*; *Felleman and Van Essen, 1991*). Consistent with the laminar response profiles, cortical depth-dependent informational connectivity analyses showed that both feedforward and feedback signals play important roles in color and disparity processing in the ventral (i.e., V2-V4) and dorsal (i.e., V2-V3ab) visual streams, respectively.

A previous electrophysiological study investigated laminar neural activity in V1 and V2 to naturalistic textures in anesthetized macaques (*Ziemba et al., 2019*). Their findings suggest that the superficial and deep layers of V1 are subject to top-down modulation from higher-order visual areas during processing of naturalistic textures. Consistent with this study, we found the strongest selectivity to naturalistic textures in the deep V1, suggesting feedback modulation from higher-order visual areas. V1 responses to naturalistic textures were also significantly weaker compared to spectrally matched noise, consistent with the framework of predictive coding: top-down hypotheses from higher-level area 'explain away' or reduce the prediction error signals in the lower visual area (*Friston, 2005*; *Murray et al., 2002*; *Rao and Ballard, 1999*). This could also explain the strongest signal reduction in the superficial cortical depth in our results since the error signals should be mainly represented in the output layers according to the model of canonical microcircuits for predictive coding (*Bastos et al., 2012*).

In V2, *Ziemba et al., 2019* found stronger texture selectivity in the superficial and middle layers, which potentially emerged from local processing within this area. In contrast, our data revealed the highest selectivity to naturalistic textures in the deep V2, along with a significant feedback modulation from V4. The amount of texture selectivity in V4 was also stronger compared to V2. This is consistent with previous macaque studies showing stronger texture selectivity in V4 than in V2 (*Okazawa et al., 2017*) and local clustered neurons in V4 that shared preferred image statistics (*Hatanaka et al., 2022*; *Kim et al., 2022*). In addition, our pilot analyses also suggest modular organizations for naturalistic texture, in addition to those for color and disparity in V4 and V3ab (*Figure 2—figure supplement 8*). Altogether, these findings suggest an important role of feedback modulation from V4 in generating selectivity to naturalistic textures in V2. In the monkey study, it is possible that general anesthesia substantially reduced feedback modulation from high-order brain areas. There is evidence showing that V4 activity is more closely related to conscious visual perception than the early visual areas (*Mehta et al., 2000*; *Tong, 2003*).

Our fMRI data demonstrate a significant contribution of feedback signals in generating the selectivity for naturalistic textures in area V2. Nonetheless, this does not preclude the possibility that such selectivity might arise from local processing within this area and become progressively stronger along the feedforward pathway. In accordance with the predictive coding hypothesis discussed above, top-down feedback might reduce the neural activity representing prediction errors in the superficial layers of lower-order areas, counteracting the effect of neural response evoked by local processing. Furthermore, recent macaque studies have shown that the cortical processing of naturalistic textures depends on the type of statistical regularities (*Kim et al., 2022*; *Okazawa et al., 2015*). Compared to previous studies (*Freeman et al., 2013*; *Okazawa et al., 2015*; *Ziemba et al., 2016*), our V2 results showed much weaker selectivity to naturalistic textures. This could be due to different textures used in the current study. As suggested by the previous findings, texture selectivity across neurons in V2 and V4 can be highly diversified (Figure 2e in *Freeman et al., 2013*; *Kim et al., 2022*): some texture families are much more or less effective in driving neural activity than others, with distinct temporal dynamics.

Our data did not show clear evidence of a 'stripe'-shaped columnar organization for naturalistic textures within area V2. A cortical column is formed by many mini-columns bound together by short-range horizontal connections (*Mountcastle, 1997*), supporting efficient information processing via local circuitry. Thus, the absence of a columnar organization in area V2 is consistent with a dominant role of feedback modulation, rather than local or feedforward processing in generating texture selectivity within this area. Considering the complex computations required for processing naturalistic information, it is likely that V4 neurons are more suitable for this task than those in V2. Higher-order statistics in naturalistic textures are computed via integrating local elements across different locations, orientations, and spatial scales (*Portilla and Simoncelli, 2000*), presenting a challenge for an early visual area such as V2 to develop a specialized computational module. In line with this idea, the neural tunings in V4 are distributed in a way suitable for categorizing textures and predicting texture discrimination abilities (*Hatanaka et al., 2022*; *Kim et al., 2022*; *Okazawa et al., 2015*). Consistently, our results also revealed modular organizations for textures in V4 and V3ab (*Figure 2—figure supplement 8*). These texture-selective organizations may be related to surface representations in these higher-order visual areas (*Wang et al., 2024*). Although cortical columns in V2 are much larger than those in V1 (*Lu and Roe, 2007*), our results do not rule out the possibility that smaller modules

of texture processing might exist beyond our fMRI resolution at 1 mm isotropic voxels, especially at farther eccentricities (*Figure 2A and D*).

Finally, the critical period for the formation of cortical columns in lower-level visual areas might close at an earlier stage during development (*Kiorpes, 2015*; *Levi, 2005*). It is possible that the emergence of selectivity to naturalistic textures requires extensive visual experience with the ability to actively explore the natural environment. After the closure of the critical period in V2 for forming color- and disparity-selective columns, V4 may still be in its critical period with high neural plasticity, allowing it to develop neuronal clusters with strong preference for naturalistic textures (*Hatanaka et al., 2022*). Subsequently, feedback modulations from V4 may further increase the selectivity for naturalistic textures in V2.

In summary, the present study demonstrated parallel pathways for color, disparity, and texture processing in the human visual cortex. Unlike color and disparity, no clear evidence of columnar organization or response preference across thick, pale, and thin stripes was found for naturalistic texture in area V2. Consistent with this finding, our results further suggest that feedback processing from V4 plays a dominant role in generating texture selectivity within V2. These results underscore the critical involvement of higher-order visual areas in texture processing. Given the diversity of naturalistic textures, different cortical mechanisms may be involved at various processing stages (*Kim et al., 2022*; *Okazawa et al., 2015*). In future studies, it would be important to characterize columnar and laminar fMRI responses using different texture types to obtain a comprehensive picture of naturalistic texture processing along the visual hierarchy.

Due to the limitations of the T2*w GE-BOLD signal in its sensitivity to large draining veins (*Fracasso et al., 2021*; *Parkes et al., 2005*; *Uludag and Havlicek, 2021*), the original BOLD responses were strongly biased toward the superficial depth in our data (*Figure 3—figure supplement 1*). Compared to GE-BOLD, VASO-CBV and SE-BOLD fMRI techniques have higher spatial specificity but much lower sensitivity (*Huber et al., 2019*). As suggested by previous studies (*Kim and Fukuda, 2008*; *Qian et al., 2024*; *Yacoub et al., 2008*), differential BOLD signals in a continuous stimulus design exhibit stronger microvascular contribution, which might help improve the laminar specificity of feature selectivity measures in our results (*Figure 3*). Compared to the submillimeter voxels, as used in most laminar fMRI studies, our fMRI resolution at 1 mm isotropic voxel may have a stronger partial volume effect in the cortical depth-dependent analysis. However, consistent with our results, previous studies have also shown that 7T fMRI at 1 mm isotropic resolution can resolve cortical depth-dependent signals in human visual cortex (*Roefs et al., 2024*; *Shao et al., 2021*). Finally, the interpretations of cortical depth-dependent results are based on the canonical models of cortical microcircuitry. Alternative connections exist in addition to these major pathways. Therefore, our findings on the cortical microcircuitry in humans would require further support from invasive neuroscience methods, such as laminar electrophysiological recordings in non-human primates.

# Materials and methods

**Key resources table**

| Reagent type (species) or resource | Designation | Source or reference | Identifiers | Additional information |
|---|---|---|---|---|
| Software, algorithm | FreeSurfer (version 6.0) | https://surfer.nmr.mgh.harvard.edu | RRID:SCR_001847 | |
| Software, algorithm | AFNI | http://afni.nimh.nih.gov/afni/ | RRID:SCR_005927 | |
| Software, algorithm | ANTs | *Avants et al., 2011* | RRID:SCR_004757 | |
| Software, algorithm | mripy | https://pypi.org/project/mripy/; *herrlich10, 2025* | | |
| Software, algorithm | MATLAB | MathWorks | RRID:SCR_001622 | |
| Software, algorithm | Psychophysics toolbox | http://psychtoolbox.org/ | RRID:SCR_002881 | |
| Software, algorithm | JASP | https://jasp-stats.org/ | RRID:SCR_015823 | |
| Software, algorithm | Portilla-Simoncelli model | *Portilla and Simoncelli, 2000* | - | Used for synthesizing naturalistic texture |

*Continued on next page*

*Continued*

| Reagent type (species) or resource | Designation | Source or reference | Identifiers | Additional information |
|---|---|---|---|---|
| Other | 7T MAGNETOM MRI scanner | Siemens Healthineers | - | MRI data collection |
| Other | 32-channel receive, 4-channel transmit open-face surface coil | *Sengupta et al., 2016* | - | Custom-built open-face visual coil |
| Other | Custom anaglyph spectacles (red and cyan) | This paper | - | Used for disparity experiment |

## Participants

Ten participants (four females; age range 21–40 years) were recruited for this study. All participants had normal or corrected-to-normal vision and reported no history of neuropsychological or visual disorders. A sample size of 10 participants is relatively large according to the literature on columnar and laminar fMRI studies with long scanning durations (*Nasr et al., 2016*; *Tootell and Nasr, 2017*). The experimental procedures were approved by the ethical review board of Institute of Biophysics, Chinese Academy of Sciences (no. 2012-IRB-011). Written informed consent was obtained from all participants prior to their participation in the study.

## General procedures

Each participant underwent three fMRI experiments in the 7T scanner. Six subjects participated in color, disparity, and texture experiments in three daily sessions (five subjects scanned both color and disparity experiments in two sessions, and the texture experiment in a single session; one subject scanned one experiment in each session). For each experiment, 10 runs of fMRI data were collected. The remaining four participants completed all three experiments in a single session, consisting of 12 runs in total (four runs for each experiment). The order of stimulus presentation was counterbalanced both within and across fMRI runs for each participant. Visual stimuli subtended 46.7° × 35.9° in visual angle, with a fixation point (0.3° in diameter) in the center. During fMRI scans, each run started and ended with 16 s fixation periods. In the remaining periods, visual stimuli were presented in 24 s stimulus blocks. Participants were required to maintain fixation and to detect sparsely and randomly presented color changes of the fixation point.

## Stimuli and apparatus

Visual stimuli were presented through an MRI-safe projector (1024 × 768 pixel resolution, 60 Hz refresh rate) onto a rear-projection screen. The experiment was conducted using MATLAB 2016a (MathWorks) based on Psychophysics toolbox extensions version 3.0 (*Brainard, 1997*; *Pelli, 1997*). Participants viewed the screen via a mirror mounted inside the head coil.

### Color experiment

The MRI-safe projector was calibrated using a PR-655 photometer to have a linear luminance output. To account for changes in isoluminance at different eccentricities (*Nasr et al., 2016*; *Bilodeau and Faubert, 1997*; *Livingstone and Hubel, 1987*; *Mullen, 1985*), we measured blue-red and blue-gray isoluminance for each participant at three eccentricity ranges (0°–3°, 3°–8°, and 8°–16°). Blue was set as the reference color because the project has lower light intensity for blue compared with red and gray. A minimal motion procedure was used to match the perceived luminance between blue and red/gray (*Anstis and Cavanagh, 1983*). During isoluminance adjustment, achromatic and chromatic gratings were presented in alternating frames, with pi/2 phase difference between adjacent frames. Blue luminance was fixed at the maximum level, participants adjusted the match-color luminance until no consistent apparent motion was seen (i.e., bi-stable motion directions with equal durations). For each eccentricity range, the isoluminance adjustment was repeated four times and the results were averaged. *Figure 1—figure supplement 1A and B* illustrates the blue-matched luminance levels (in RGB index) of gray and red, respectively, at the three eccentricity ranges. Consistent with previous findings (*Nasr et al., 2016*; *Bilodeau and Faubert, 1997*; *Livingstone and Hubel, 1987*; *Mullen, 1985*), the isoluminance level varied significantly as eccentricity (blue-gray: $F(2,9) = 87.9$, p<0.001, FWE corrected; blue-red: $F(2,9) = 35.71$, p<0.001, FWE corrected). During fMRI scans, chromatic

and achromatic gratings (0.2 cycles-per-degree concentric rings, 46.7° × 35.9° in size, *Figure 1A*, left panel) moved in either centrifugal or centripetal direction with a speed of 0.8 cycles/s, alternating in 24 s blocks (five blocks per stimulus condition). A 16 s fixation period with uniform gray background was presented at the beginning and the end of each run. Ten runs of fMRI data were collected for six subjects and four runs for the other four subjects.

### Disparity experiment

The binocular disparity stimulus (46.7° × 35.9°) was random red/green dot stereograms (RDSs) presented against a black background. Subjects viewed the stimulus through custom anaglyph spectacles (red and cyan). The RDSs generated a stereoscopic percept, with the depth of each dot sinusoidally modulating between –2.2°–2.2° in front of and behind the frontoparallel plane of fixation (*Figure 1A*, middle panel). In the zero-disparity 2D control condition, randomly moving dots formed a frontoparallel plane intersecting the fixation point (i.e., zero depth at that point). Each run began and ended with a 16 s fixation period. The disparity-defined grating and zero-disparity disc stimuli were presented in alternation every 24 s. Ten runs of fMRI data were collected for six subjects and four runs for the other four subjects.

### Texture experiment

The naturalistic texture and spectrally matched noise (*Figure 1A*, right panel) were synthesized using the Portilla–Simoncelli model (*Portilla and Simoncelli, 2000*). Thirty image pairs were generated. The stimuli (46.7° × 35.9°) were presented in the middle of the screen, centered on the fixation point. Each fMRI run consisted of ten 24 s stimulus blocks, starting and ending with 16 s fixation periods. Each stimulus block consisted of 30 pictures in a random order, with a duration of 0.6 s for each picture. Ten runs of fMRI data were collected for six subjects and four runs for the other four subjects. Although the number of runs was not equal across participants, there were at least four runs (i.e., 20 blocks for each stimulus condition) of data in each experiment, which should be sufficient to investigate within-subject effects. In support of this, the split-half analysis revealed reproducible columnar organizations with five runs of data (*Figure 2*).

## MRI data acquisition

MRI data were acquired on a 7T MAGNETOM MRI scanner (Siemens Healthineers, Erlangen, Germany) with a custom-built 32-channel receive and 4-channel transmit open-face surface coil (*Sengupta et al., 2016*), in the Beijing MRI center for Brain Research. Functional data were collected with a T2*-weighted 2D GE-EPI sequence (1.0 mm isotropic voxels, 39 slices, TR = 2400 ms, TE = 25 ms, image matrix = 128 × 128, FOV = 128 × 128 mm², GRAPPA acceleration factor = 3, nominal flip angle = 80°, partial Fourier factor = 7/8, phase encoding direction from head to foot, receiver bandwidth = 1148 Hz/Pix). Slices were oriented perpendicular to the calcarine sulcus. After each fMRI run, five EPI images with reversed phase encoding direction (F to H) were also acquired for EPI distortion correction. High-resolution anatomical volumes were acquired with a T1-weighted MP2RAGE sequence at 0.7 mm isotropic resolution (256 sagittal slices, centric phase encoding, acquisition matrix = 320 × 320, FOV = 224 × 224 mm², GRAPPA = 3, TR = 4000 ms, TE = 3.05 ms, TI1=750 ms, flip angle = 4°, TI2 = 2500 ms, flip angle = 5°). A bite-bar was settled for each subject to minimize head motion to ensure high data quality.

## MRI data analysis

### Preprocessing

The anatomical data were preprocessed using FreeSurfer version 6.0 (*Fischl, 2012*), which involved the segmentation and reconstruction of inflated and flattened cortical surfaces based on high-resolution anatomical data. We inspected visually and edited manually the surface segmentation to eliminate dura matter, sinus, etc., ensuring correct gray matter boundaries. The functional data were preprocessed and analyzed with AFNI (*Cox, 1996*), ANTs (*Avants et al., 2011*), and the mripy package developed in our lab (https://github.com/herrlich10/mripy; *herrlich10, 2025*). Preprocessing steps included head motion correction, de-spiking, slice timing correction, EPI distortion correction (nonlinear warping with blip-up/down method), and per-run scaling as percent signal change. All spatial transformations were combined and applied in a single interpolation step (sinc interpolation)

to minimize the loss of spatial resolution (*Wang et al., 2022*). No spatial smoothing was applied to the main functional imaging data. We aligned the anatomical volume as well as the reconstructed surfaces to the mean of preprocessed EPI images. General linear models (GLMs) were used to estimate the BOLD responses (β values) to visual stimuli with a canonical hemodynamic response function (BLOCK4 in AFNI). Slow baseline drift and motion parameters were included as nuisance regressors in GLMs.

## Cortical depth definition

To perform the cortical depth-dependent analysis, we resampled the functional volumes to 0.5 mm isotropic resolution using cubic interpolation (3dresample in AFNI). An equi-volume method was used to calculate the relative cortical depth of each voxel to the white matter and pial surface (0: white matter surface; 1: pial surface, *Figure 3—figure supplement 2A*), using mripy (https://github.com/herrlich10/mripy). The voxels in each ROI were sorted and divided into three bins: deep depth (0–0.33), middle depth (0.33–0.67), and superficial depth (0.67–1.00) (*Ge et al., 2020*; *Kemper et al., 2018*).

## ROI definition

ROIs were defined on the inflated cortical surface. Surface ROIs for V1, V2, V3ab, and V4 were defined based on the polar angle atlas from the 7T retinotopic dataset of Human Connectome Project (*Benson et al., 2014*; *Benson et al., 2018*). Moreover, the boundary of V2 was edited manually based on columnar patterns. All ROIs were constrained to regions where mean activation across all stimulus conditions exceeded 0. In V2, ROIs for the thin and thick 'stripe'-shaped columns were manually defined in two stages. Firstly, we defined thin stripes by contrast between the chromatic and achromatic stimuli, and thick stripes by contrast between binocular disparity and 2D control stimuli. Secondly, we defined final stripes by contrast between these two, resulting in interdigitated thin and thick stripes distributed without overlap. The pale stripes were defined as the regions located between the thin and thick stripes. We compared the fMRI signal changes elicited by the three stimulus contrasts in each stripe. For the cortical depth-dependent analyses in *Figure 3*, we used all voxels in the retinotopic ROI. Pooling all voxels in the ROI avoids the problem of double-dipping and also increases the signal-to-noise ratio of ROI-averaged BOLD responses.

## Stimulus-selectivity index

The ROI-averaged BOLD responses were calculated for each stimulus condition. We defined a selectivity index (SI) for color, disparity, and texture processing, respectively:

$$\text{SI}_{\text{color}} = (\beta_{\text{char}} - \beta_{\text{ach}}) / (\beta_{\text{char}+\beta_{\text{ach}}})$$
$$\text{SI}_{\text{disparity}} = (\beta_{\text{3D}} - \beta_{\text{2D}}) / (\beta_{\text{3D}} + \beta_{\text{2D}})$$
$$SI_{texture} = (\beta_T - \beta_N) / (\beta_T + \beta_N)$$

Here, $\beta_{\text{chr}}$, $\beta_{\text{ach}}$, $\beta_{\text{3D}}$, $\beta_{\text{2D}}$, $\beta_T$, and $\beta_N$ represent the beta estimates of BOLD responses for the chromatic, achromatic, binocular disparity, 2D control, naturalistic texture, and spectrally matched noise stimuli, respectively.

## Test–retest reliability of columnar organizations

For five subjects who participated in both color and disparity experiments across two daily scan sessions, we generated color ($\beta_{\text{chr}} - \beta_{\text{ach}}$) and disparity ($\beta_{\text{3D}} - \beta_{\text{2D}}$) selective functional maps on the cortical surface in area V2. Pearson's correlations were computed to evaluate the test–retest reliability of color- and disparity-selective response patterns between the two scan sessions. FWEs of the pattern correlations were controlled by a null distribution generated from Monte Carlo simulation (*Qian et al., 2023*). In this procedure, the first session's GLM residual volumes were used to estimate the spatial auto-correlation function (3dFWHMx in AFNI), and it was then used to generate a simulated GLM volume for the second session (3dClustSim in AFNI). We then projected the first and second sessions' GLM volumes onto the cortical surface and calculated the inter-session correlations of color- and disparity-selective response patterns in V2. This process was repeated 10,000 times

(*Figure 2—figure supplement 9*). Finally, the measured correlation coefficients were compared to the critical value of the null distribution.

## Pial vein removal

To mitigate the strong BOLD effect from large pial veins on layer-specific signals in the gray matter (*Cheng et al., 2001*; *Gati et al., 1997*; *Kay et al., 2019*; *Yacoub et al., 2005*), we excluded vertices with extremely large signal changes and their corresponding voxels in the gray matter. Specifically, the top 5% cortical vertices with large signal changes from baseline (all stimulus conditions vs. fixation) and the corresponding voxels across all cortical depths were excluded from analysis in V1, V2, V4, and V3ab (*Figure 3—figure supplement 2B*). According to our previous study (*Liu et al., 2021*), this large vein removal approach can effectively reduce the superficial bias in laminar response profiles of the visual cortex.

## Cortical depth-dependent informational connectivity

To investigate stimulus-specific information flow in the visual processing hierarchy, we calculated informational connectivity between the input and output layers of two brain regions (*Aly and Turk-Browne, 2016*; *Coutanche and Thompson-Schill, 2014*; *Haxby et al., 2001*; *Huffman and Stark, 2017*; *Jia et al., 2020*; *Koster et al., 2018*). This approach is analogous to the functional connectivity method in multivariate pattern analysis, where connectivity is inferred from shared changes (covariation) in decoding accuracy between regions over time (i.e., across blocks). The layer-specific neural circuitry was further used to define the direction of informational connectivity (*Jia et al., 2020*). Specifically, feedforward connectivity was defined as the connection between the superficial layer of the lower visual area and the middle layer of the higher visual area, whereas feedback connectivity was defined as the connection between two deep layers (*Figure 1C*).

In the analysis, a separate GLM regressor was first used to estimate the activation pattern (*t* scores of voxels) for each stimulus block. For each stimulus condition per cortical layer, 50 activation patterns were obtained for six subjects and 20 activation patterns for the other four subjects. We trained linear SVM classifiers (https://www.csie.ntu.edu.tw/~cjlin/libsvm/) using these patterns and extracted the distance from the hyperplane for each stimulus block, following a leave-one-run-out cross-validation procedure. Before each training, feature selection was performed with K-1 runs of data to select voxels with high visual sensitivity and stimulus selectivity. We first selected the top 20% most visually responsive voxels by the *t* distribution of activations from baseline for a stimulus condition (e.g., Chromatic + Achromatic – Fixation). Then 200 voxels with strong stimulus selectivity were selected from each side of the *t* distribution of differential responses (e.g., Chromatic – Achromatic). The activation patterns of these 400 voxels were normalized to have a unitary Euclidean norm (L2-norm). An SVM classifier of stimulus type (e.g., Chromatic vs. Achromatic) was trained from K-1 runs of data, and the distance to the decision boundary was calculated for each stimulus block from the remaining run. Pearson's correlation between the block-by-block distance timeseries of two brain regions was calculated to estimate the layer-specific informational connectivity. Then we averaged the correlation coefficients across all folds.

The correlation coefficients were Fisher z-transformed before statistical analysis. One-sample *t*-test against 0 was conducted on each connectivity value, and the results were submitted to false discovery rate (FDR) correction. The reported correlations are the original *r* values to facilitate interpretation and visualization.

## Statistical analysis

Statistical analyses were performed using MATLAB 2021a, JASP (v0.17.1), and custom Python code. Repeated-measures ANOVA and paired *t*-tests were used for most of the statistical analyses of ROI data. An FWE-corrected threshold of p<0.05 was used for each group of ANOVA. We further performed an FWE correction for paired *t*-tests only when the corrected p-value from ANOVA exceeded threshold, according to the (*Fisher, 1936*; *Levin et al., 1994*). We conducted Bayesian repeated-measures ANOVA to complement the classical null-hypothesis test with JASP (*Wagenmakers et al., 2018*). The calculated Bayes factor ($BF_{10}$) falling into 0.33–1 indicates anecdotal evidence for the null hypothesis ($H_0$), whereas a value between 10–30 and >100 refers to strong and extreme evidence for the alternative hypothesis ($H_1$), respectively. Moreover, we used the nonparametric permutation test to test if

selectivity indices differ across stripes. For each selectivity index, we resampled with replacement and computed the mean value within each type of stripe, and calculated the difference between each pair of stripes. This process was repeated 10,000 times to derive the null distribution. The critical value was set to 0.

## Acknowledgements

This study was funded by STI2030-Major Projects (2022ZD0211900, 2021ZD0203600, 2021ZD0204200), National Natural Science Foundation of China (31971031, 31871107, 31930053), and Hunan Provincial Natural Science Foundation (2024JJ6313).

## Additional information

### Funding

| Funder | Grant reference number | Author |
|---|---|---|
| National Science and Technology Innovation STI2030-Major Projects | 2022ZD0211900 | Peng Zhang |
| National Science and Technology Innovation STI2030-Major Projects | 2021ZD0203600 | Nihong Chen |
| National Science and Technology Innovation STI2030-Major Projects | 2021ZD0204200 | Peng Zhang |
| National Natural Science Foundation of China | 31971031 | Nihong Chen |
| National Natural Science Foundation of China | 31871107 | Peng Zhang |
| National Natural Science Foundation of China | 31930053 | Nihong Chen Peng Zhang |
| Hunan Provincial Natural Science Foundation | 2024JJ6313 | Chengwen Liu |

The funders had no role in study design, data collection and interpretation, or the decision to submit the work for publication.

### Author contributions

Hailin Ai, Conceptualization, Resources, Data curation, Software, Formal analysis, Validation, Investigation, Visualization, Methodology, Writing – original draft, Project administration, Writing – review and editing; Weiru Lin, Data curation, Software, Formal analysis, Supervision, Validation, Investigation, Visualization, Methodology; Chengwen Liu, Data curation, Software, Formal analysis, Supervision, Investigation, Methodology; Nihong Chen, Peng Zhang, Conceptualization, Resources, Data curation, Software, Formal analysis, Supervision, Funding acquisition, Validation, Investigation, Visualization, Methodology, Writing – original draft, Project administration, Writing – review and editing

### Author ORCIDs

Nihong Chen  https://orcid.org/0000-0002-0890-3875
Peng Zhang  https://orcid.org/0000-0002-9603-8454

### Ethics

Human subjects: The experimental procedures were approved by the ethical review board of Institute of Biophysics, Chinese Academy of Sciences (No. 2012-IRB-011). Written informed consent was obtained from all participants prior to their participation in the study.

Reviewer #1 (Public review): https://doi.org/10.7554/eLife.93171.3.sa1

Reviewer #2 (Public review): https://doi.org/10.7554/eLife.93171.3.sa2
Reviewer #3 (Public review): https://doi.org/10.7554/eLife.93171.3.sa3
Author response https://doi.org/10.7554/eLife.93171.3.sa4

## Additional files

### Supplementary files
MDAR checklist

### Data availability
Data and code to reproduce the main findings of this study can be downloaded from Open Science Framework (OSF, https://doi.org/10.17605/OSF.IO/VBTQS).

The following dataset was generated:

| Author(s) | Year | Dataset title | Dataset URL | Database and Identifier |
|---|---|---|---|---|
| Ai H, Lin W, Liu C, Chen N, Zhang P | 2024 | Mesoscale functional organization and connectivity of color, disparity, and naturalistic texture in human second visual area | https://doi.org/10.17605/OSF.IO/VBTQS | Open Science Framework, 10.17605/OSF.IO/VBTQS |

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
