## [Editor Report · eLife Assessment]

This study builds on previous findings showing modular organization of primate visual cortical areas by presenting **important** results about the cortical processing of color, disparity, and naturalistic textures in the human visual cortex at the spatial scale of cortical layers and columns using state-of-the-art high-resolution fMRI methods at ultra-high magnetic field strength (7T). **Solid** evidence supports an interesting layer-specific informational connectivity analysis to infer information flow across early visual areas for processing disparity and color signals. While the question of how the modularity of representation relates to cortical hierarchical processing is interesting, the findings that texture does not map onto previously established columnar architecture in V2 is suggestive. The successful application of high-resolution fMRI methods to study the functional organization along cortical columns and layers is relevant to a broad readership interested in general neuroscience.

---

## [Referee Report · Reviewer #1 (Public review)]

Summary:

This study examines the cortical modular functional organization of visual texture in comparison with that of color and disparity. While color, disparity, and orientation have been shown to exhibit clear functional organizations within the thin, thick, and thick/pale stripes of V2, whether the feature of texture is also organized within V2 is unknown. Using ultrahigh field 7T fMRI in humans viewing color-, disparity-, and texture-specific visual stimuli, the authors find that, unlike color and disparity, texture does not exhibit stripe-specific organization in V2. Moreover, using laminar imaging methods and calculations of informational connectivity, they find V2 color and disparity stripes exhibit the expected feedforward and feedback relationships with V1 & V4, and with V1 & V3ab, respectively. In contrast, texture activation, found predominantly in the deep layers of V2, is driven preferentially by feedback from V4. Based on these findings, the authors suggest that texture is a visual feature computed in higher-order areas and not generated by local intra-V2 computation.

Strengths:

This study poses an interesting and fundamental question regarding the relationship between functional modularity and hierarchical origin of computed properties. This question is thus highly significant and deserves study. The methodology is appropriate for the question and the areal and laminar resolution achieved across 10 subjects is commendable. The combination of high-resolution functional imaging and informational connectivity analysis introduces a useful way for examining feedforward and feedback relationships in mesoscale imaging data.

Comments on latest version:

The authors have responded adequately to my comments. The lack of texture organization in V2 is now strengthened by the apparently more clustered texture response in V4 (Fig. S9). The paired results in V2 and V4 make the study stronger. The authors may suggest that texture response, while present at the neural level, may not emerge as a primary organizational cue in V2, based on this texture stimulus paradigm. The negative results should still be presented cautiously. The connectivity inferences are interesting but should also be stated cautiously, as there are multiple assumptions. Overall, this study makes a contribution to emerging views about texture processing in the early visual pathways.

---

## [Referee Report · Reviewer #2 (Public review)]

This study investigates the cortical circuitry at the mesoscopic level of cortical columns in the human secondary visual cortex (V2) using high-resolution fMRI at ultra-high field strength (7T). The findings confirm the columnar organization of color-selective thin and disparity-selective thick stripes, a result previously demonstrated and replicated in human fMRI research. However, this study adds a novel layer of analysis by examining cortical depth, providing insights into feedforward and feedback connections to and from V2. Furthermore, examining texture selectivity in V2 showed no evidence of a columnar structure when compared to color- and disparity-selective activation clusters. Interestingly, texture selectivity in V2 was most pronounced in deeper cortical layers, with significant feedback connectivity from V4. The authors conclude that local columnar circuitry plays a crucial role in color and disparity processing within V2, while texture selectivity is driven by feedback modulation. This research underscores the potential of high-resolution human fMRI to explore the local circuitry of the cortex at the mesoscopic scale.

However, I still have a few comments that I would like to be addressed:

(1) In lines 401-403, the authors state that differential BOLD responses can significantly enhance the laminar specificity. Differential contrasts indeed have the potential to reduce macrovascular contributions that are unspecific to both experimental conditions, which was already discussed in the literature (e.g., Yacoub et al., 2008, High-field fMRI unveils orientation columns in humans). This might be especially true for the pial vasculature that drains a larger surface area of the cortex, e.g., multiple columns, which is probably the key factor that enables cortical column mapping using differential BOLD contrasts despite the relatively large spatial point spread function of the BOLD response. However, this may differ for laminar analyses, where neuronal and vascular responses from intracortical and pial veins might be harder to disentangle. It would, therefore, be advisable to tone down this statement somewhat since it could imply that laminar specificity can be readily achieved with GE-BOLD, while this remains an active area of research. This is not to say that the present results are incorrect, but the broader implications of this statement should be cautiously framed.

(2) Looking at Figure 3, one might also argue (excluding responses from V4) that statistically significant differences in selectivity are only observed where the cortical profiles generally show higher response levels. Could this be simply due to varying signal-to-noise ratios (SNR) achieved by different contrasts (color, disparity, texture)?

(3) In lines 480-484, the authors state that twenty blocks for each stimulus condition should be sufficient to investigate within-subject effects. It would be helpful if they could elaborate on the basis for this claim. High-resolution fMRI is typically limited by low temporal signal-to-noise ratio (tSNR), and extensive averaging is often required to achieve sufficient signal. Clarifying the rationale behind this assertion would strengthen the argument.

---

## [Referee Report · Reviewer #3 (Public review)]

Summary:

Ai et al. studied texture, color and disparity selectivity in human visual cortex at mesoscale level using high-resolution fMRI. They reproduced earlier monkey and human studies showing interdigitated color-selective and disparity-selective sub-compartments within area V2, likely corresponding to thin and thick stripes, respectively. At least with the stimuli used, no clear evidence for texture-selective mesoscale activations were observed in area V2. The most interesting and novel part of this study focused on cortical-depth-dependent connectivity analyses across areas. The data suggest feedback and feedforward functional connectivity between V1 and V3A for disparity signals and feedback from V4 to the deep layers of V2 for textures.

Strengths:

High-resolution fMRI and highly interesting layer-specific informational connectivity analyses.

Weaknesses:

The authors tend to overclaim their results. Too few data to make conclusive inferences.

---

## [Author Response]

The following is the authors’ response to the original reviews.

**Public Review:**

**Reviewer #1:**
(1) To support the finding that texture is not represented in a modular fashion, additional possibilities must be considered. These include (a) the effectiveness and specificity of the texture stimulus and control stimuli, (b) further analysis of possible structure in images that may have been missed, and (c) limitations of imaging resolution.

Thank you for your comments. To address your concerns, we have conducted a new 3T fMRI experiment to demonstrate the effectiveness and specificity of our stimuli, performed further analyses to investigate possible structure of texture-selective activation, and discussed the limitations of imaging resolution.

(a) To demonstrate the effectiveness and specificity of our stimuli, we conducted a new 3T fMRI experiment in five participants using an experimental design and texture families similar to those in Freeman (2013). Six texture stimuli in the 7T experiment were also included. To assess the effectiveness of each stimulus type, different texture families and their corresponding noise patterns were presented in separate blocks for 24 seconds, at a high presentation rate of 5 frames per second. In Figure S7, all texture families showed significantly stronger activation in V2 compared to their corresponding noise patterns, even for those that ‘appeared’ to have residual texture (e.g., the third texture family). These results demonstrate that our texture vs. noise stimuli were effective in producing texture-selective activations in area V2. Compared to the 7T results, the 3T data showed a notable increase in texture-selective activations in V2, likely due to increased stimulus presentation speed (1.25 vs. 5 frames/second). Future studies should use stimuli with faster presentation speed to validate our results in the 7T experiment.

(b)Thank you for pointing out the possible structures of texture-selective activations in the peripheral visual field (Figure S1). In further analyses, we also found stronger texture selectivity in more peripheral visual fields (Figure 2D), and there were weak but significant correlations in the texture-noise activation patterns during split-half analysis (Author response image 2). Although this is not strong evidence for columnar organization of naturalistic textures, it suggests a possibility for modular organizations in the peripheral visual field.

(c) Although our fMRI result at 1-mm isotropic resolution did not show strong evidence for modular processing of naturalistic texture in V2 stripe columns, this does not exclude the possibility that smaller modules exist beyond the current fMRI resolution. We have discussed this possibility in the revised manuscript.

We hope this response clarifies our findings, and we have revised the conclusions in the manuscript accordingly.

(2) More in-depth analysis of subject data is needed. The apparent structure in the texture images in peripheral fields of some subjects calls for more detailed analysis. e.g Relationship to eccentricity and the need for a 'modularity index' to quantify the degree of modularity. A possible relationship to eccentricity should also be considered.

Based on your recommendations, we have performed further analysis and found interesting results regarding the modularity index in relation to eccentricity. As shown in Figure 2D, the texture-selectivity index increased as eccentricity. This may suggest a higher possibility of modular organization for texture representation in the peripheral compared to central visual fields. We have updated our results in Figure 2C, and discussed this possibility in the revised manuscript.

(3) Given what is known as a modular organization in V4 and V3 (e.g. for color, orientation, curvature), did images reveal these organizations? If so, connectivity analysis would be improved based on such ROIs. This would further strengthen the hierarchical scheme.

Following your recommendations, we have conducted further analysis to investigate the potential modular organizations in V4 and V3ab. In Figure S9 (Figure S9), vertices that are most responsive to color, disparity and texture were shown in a representative subject. Indeed, texture-selective patches can be found in both V4 and V3ab, along with the color- and disparity-selective patches. We agree with you that there should be pathway-specific connectivity among the same type of functional modules. In the informational connectivity analyses, we already used highly informative voxels by feature selection, which should mainly represent information from the modular organizations in these higher visual areas.

**Reviewer #2:**
(1) In lines 162-163, it is stated that no clear columnar organization exists for naturalistic texture processing in V2. In my opinion, this should be rephrased. As far as I understand, Figure 2B refers to the analysis used to support the conclusion. The left and middle bar plots only show a circular analysis since ROIs were based on the color and disparity contrast used to define thin and thick stripes. The interesting graph is the right plot, which shows no statistically significant overlap of texture processing with thin, thick, and pale stripe ROIs. It should be pointed out that this analysis does not dismiss a columnar organization per se but instead only supports the conclusion of no coincidence with the CO-stripe architecture.

Thank you for your suggestions. Reviewer #1 also raised a similar concern. We agree that there may be a smaller functional module of textures in area V2 at a finer spatial scale than our fMRI resolution. We have rephrased our conclusions to be more precise.

(2) In Figure 3, cortical depth-dependent analyses are presented for color, disparity, and texture processing. I acknowledge that the authors took care of venous effects by excluding outlier voxels. However, the GE-BOLD signal at high magnetic fields is still biased to extravascular contributions from around larger veins. Therefore, the highest color selectivity in superficial layers might also result from the bias to draining veins and might not be of neuronal origin. Furthermore, it is interesting that cortical profiles with the highest selectivity in superficial layers show overall higher selectivity across cortical depth. Could the missing increase toward the pial surface in other profiles result from the ROI definition or overall smaller signal changes (effect size) of selected voxels? At least, a more careful interpretation and discussion would be helpful for the reader.

We agree with you that there will be residual venous effects even after removing voxels containing large veins. However, calculating the selectivity index largely removed the superficial bias (Figure 3). In the revised manuscript, we discussed the limitations of cortical depth-dependent analysis using GE-BOLD fMRI.

In Line 397-403: “Due to the limitations of the T2*w GE-BOLD signal in its sensitivity to large draining veins (Fracasso et al., 2021; Parkes et al., 2005; Uludag & Havlicek, 2021), the original BOLD responses were strongly biased towards the superficial depth in our data (Figure S8). Compared to GE-BOLD, VASO-CBV and SE-BOLD fMRI techniques have higher spatial specificity but much lower sensitivity (Huber et al., 2019). As shown in a recent study (Qian et al., 2024), using differential BOLD responses in a continuous­­ stimulus design can significantly enhance the laminar specificity of the feature selectivity measures in our results (Figure 3).”

It is unlikely that the strongest color selectivity index in the superficial depth is a result of stronger signal change or larger effect size in this condition. As shown by the original BOLD responses in Figure S8, all stimulus conditions produced robust activations that strongly biased to the superficial depth. High texture selectivity was also found in V4 and V3ab across cortical depth, which showed a flat laminar profile.

(3) I was slightly surprised that no retinotopy data was acquired. The ROI definition in the manuscript was based on a retinotopy atlas plus manual stripe segmentation of single columns. Both steps have disadvantages because they neglect individual differences and are based on subjective assessment. A few points might be worth discussing: (1) In lines 467-468, the authors state that V2 was defined based on the extent of stripes. This classical definition of area V2 was questioned by a recent publication (Nasr et al., 2016, J Neurosci, 36, 1841-1857), which showed that stripes might extend into V3. Could this have been a problem in the present analysis, e.g., in the connectivity analysis? (2) The manual segmentation depends on the chosen threshold value, which is inevitably arbitrary. Which value was used?

A previous study showed that the retinotopic atlas of early visual areas (V1-V3) aligned very well across participants on the standard surface after surface-based registration by the anatomical landmarks (Benson 2018). Thus, the group-averaged atlas should be accurate in defining the boundaries of early visual areas. To directly demonstrate the accuracy of this method, retinotopic data were acquired in five participants in a 3T fMRI experiment. A phase-encoded method was used to define the boundaries of early visual areas (black lines in Author response image 1), which were highly consistent with the Benson atlas.

Although a few feature-selective stripes may extend into V3, these stripe patterns were mainly represented in V2. Thus, the signal contribution from V3 is likely to be small and should not affect the pattern of results. The activation map threshold for manual segmentation was abs(T)>2. We have clarified this in the revised methods.

**Author response image 1. sa4fig1:** Retinotopic ROIs defined by the Benson atlas (left) and the polar angle map (right) of the representative subject. Black lines denote the boundaries of early visual areas based on the retinotopic map from the subject.

Benson, N. C., Jamison, K. W., Arcaro, M. J., Vu, A. T., Glasser, M. F., Coalson, T. S., Van Essen, D. C., Yacoub, E., Ugurbil, K., Winawer, J., & Kay, K. (2018). The Human Connectome Project 7 Tesla retinotopy dataset: Description and population receptive field analysis. J Vis, 18(13), 23. https://doi.org/10.1167/18.13.23

(4) The use of 1-mm isotropic voxels is relatively coarse for cortical depth-dependent analyses, especially in the early visual cortex, which is highly convoluted and has a small cortical thickness. For example, most layer-fMRI studies use a voxel size of around isotropic 0.8 mm, which has half the voxel volume of 1 mm isotropic voxels. With increasing voxel volume, partial volume effects become more pronounced. For example, partial volume with CSF might confound the analysis by introducing pulsatility effects.

We agree that a 1-mm isotropic voxel is much larger in volume than a 0.8-mm isotropic voxel, but the resolution along the cortical depth is not a big difference. In addition to our study, a previous study showed that fMRI at 1-mm isotropic resolution is capable of resolving cortical depth-dependent signals (Roefs et al., 2024; Shao et al., 2021). We have discussed these issues about fMRI resolution in the revised manuscript.

In Line 403-408: “Compared to the submillimeter voxels, as used in most laminar fMRI studies, our fMRI resolution at 1-mm isotropic voxel may have a stronger partial volume effect in the cortical depth-dependent analysis. However, consistent with our results, previous studies have also shown that 7T fMRI at 1-mm isotropic resolution can resolve cortical depth-dependent signals in human visual cortex (Roefs et al., 2024; Shao et al., 2021).”

Shao, X., Guo, F., Shou, Q., Wang, K., Jann, K., Yan, L., Toga, A. W., Zhang, P., & Wang, D. J. J. (2021). Laminar perfusion imaging with zoomed arterial spin labeling at 7 Tesla. NeuroImage, 245, 118724. https://doi.org/10.1016/j.neuroimage.2021.118724

Roefs, E. C., Schellekens, W., Báez-Yáñez, M. G., Bhogal, A. A., Groen, I. I., van Osch, M. J., ... & Petridou, N. (2024). The Contribution of the Vascular Architecture and Cerebrovascular Reactivity to the BOLD signal Formation across Cortical Depth. *Imaging Neuroscience, 2*, 1–19.

(5) The SVM analysis included a feature selection step stated in lines 531-533. Although this step is reasonable for the training of a machine learning classifier, it would be interesting to know if the authors think this step could have reintroduced some bias to draining vein contributions.

We excluded vertices with extremely large signal change and their corresponding voxels in the gray matter when defining ROIs. The same number of voxels were selected from each cortical depth for the SVM analysis, thus there was no bias in the number of voxels from the superficial layers susceptible to large draining veins.

**Reviewer #3:**
The authors tend to overclaim their results.

Re: Thank you for your comments. We added more control analyses to strengthen our findings, and gave more appropriate discussion of results.

**Recommendations for the authors:**

**Reviewer #1:**
(1) Controls: There is a bit more complexity than is expressed in the introduction. The authors hypothesize that the emergence of computational features such as texture may be reflected in specialized columns. That is, if texture is generated in V2, there may be texture columns (perhaps in the pale stripes of V2); but if generated at a higher level, then no texture columns would be needed. This is a very interesting and fundamental hypothesis. While there may be merit to this hypothesis, the demonstration that color and disparity are modular but not texture falls short of making a compelling argument. At a minimum, the finding that texture is not organized in V2 requires additional controls. (a) To boost the texture signal, additional texture stimuli or a sequence of multiple texture stimuli per trial could be considered. (b) Unfortunately, the comparison noise pattern also seems to contain texture; perhaps a less textured control could be designed. (c) It also appears that some of the texture images in Supplementary Figure S1 contain possible structure, e.g. in more peripheral visual fields. (d) Is it possible that the current imaging resolution is not sufficient for revealing texture domains? (e) Note that 'texture' may be a property that defines surfaces and not contours. Thus, while texture may have orientation content, its function may be associated with the surface processing pathways. A control stimulus might contain oriented elements of a texture stimulus that do not elicit texture percept; such a control might activate pale and/or thick stripes (both of which contain orientation domains), while the texture percept stimulus may activate surface-related bands in V4.

Thank you for your suggestions. They are extremely helpful in improving our manuscript. For the controls you mentioned in (a-d), we discussed them in the public review that we also attached below.

(a) and (b): To demonstrate the effectiveness and specificity of our stimuli, we conducted a new 3T fMRI experiment in five participants using an experimental design and texture families similar to those in Freeman (2013). All texture stimuli in the 7T experiment were also included. To assess the effectiveness of each stimulus type, different texture families and their corresponding noise patterns were presented in separate blocks for 24 seconds, at a high presentation rate of 5 frames per second. In Figure S7, all texture families showed significantly stronger activation in V2 compared to their corresponding noise patterns, even for those that ‘appeared’ to have residual texture (e.g., the third texture family). These results suggest that our texture stimuli were effective in producing texture-selective activations in area V2 compared to the noise control. Compared to the 7T results, the 3T data showed a notable increase in texture-selective activations in V2, likely due to the increased stimulus presentation speed (1.25 vs. 5 frames/second). Weak texture activations might preclude the detection of columnar representations in the 7T experiment.

(c) Thank you for pointing out the possible structures of texture-selective activations in the peripheral visual field (Figure S1). In further analyses, we also found stronger texture selectivity in more peripheral visual fields (Figure 2D), and there were weak but significant correlations in the texture-noise activation patterns during split-half analysis (Author response image 2). Although these are not strong evidence for columnar organization of naturalistic textures, it suggests a possibility for such organizations in the peripheral visual field.

(d) Although our fMRI result at 1-mm isotropic resolution did not show strong evidence for modular processing of naturalistic texture in V2 stripe columns, this does not exclude the possibility that smaller modules exist beyond the current fMRI resolution. We have discussed these limitations in the revised manuscript.

We fully agree with your explanation in (e). It fits our data very well. Both texture and control stimuli strongly activated the CO-stripes (Figure 2 and Figure 2D), while modular organizations for texture were found in V4 and V3ab (Figure S9). We have discussed this explanation in the revised manuscript.

In Line 371-374: “Consistently, our pilot results also revealed modular organizations for textures in V4 and V3ab (Figure S9). These texture-selective organizations may be related to surface representations in these higher order visual areas (Wang et al., 2024).”

(2) Overly simple description of FF, FB circuitry. The classic anatomical definition of feedforward is output from a 'lower' area, in most cases predominantly arising from superficial layers and projecting to middle layers of a 'higher area' (Felleman and Van Essen 1991). This description holds for V1-to-V2, V2-to-V3, and V2-to-V4. [Note there are also feedforward projections from central 5 degrees of V1-to-V4 (cf. Ungerleider) as well as V3-to-V4.] The definition of feedback can be more varied but is generally considered from cells in superficial and deep layers of 'higher' areas projecting to superficial and deep layers of 'lower' areas. Feedback inputs to V1 heavily innervate Layer 1 and superficial Layer 2, as well as the deep layers. Note that feedback connections from V2 to V1, similar to that from V1 to V2, are functionally specific, i.e. thin-to-blob and pale/thick-to interblob (Federer...Angelucci 2021, Hu...Roe 2022). Thus, current views are moving away from the dogma that feedback is diffuse. Recognition that feedback may be modular introduces new ideas about analysis.

Thanks for your detailed recommendations. We have expanded the discussion of circuit models of functional connectivity in the introduction. Our model and experiments primarily aim to investigate how higher-level areas provide feedback to the V2 area. While we acknowledge that feedback may indeed be functionally specific, our methodology has some certain advantages: it ensures signal stability and avoids the double-dipping issue. Meanwhile, it also focuses on voxels with high feature selectivity, which may already be included in the modular organizations of early visual areas. In the functional connectivity analysis, we performed feature selection to use the most informative voxels. These voxels with high feature selectivity should already be included in the modular organizations of early visual areas. Identifying functionally specific feedback connections between modular areas will be an important and meaningful work for future research. We have added a discussion of this topic in the revised manuscript.

In Line 136-138: “Only major connections were shown here. There are also other connections, such as V1 interblobs projecting to thick stripes (Federer et al., 2021; Hu & Roe, 2022; Sincich and Horton, 2005).”

(3) Imaging superficial layers: Although removal of the top layer of cortical voxels (top 5% of voxels) is a common method for dealing with surface vascular artifact contribution to BOLD signal, it likely removes a portion of the Layer 1&2 feedback signals. Is this why the authors define feedback and deep layer to deep layer? If so, both superficial and deep-layer data in Figure 4 should be explicitly explained and discussed.

Thank you for pointing this out. We would like to clarify the surface-based method removing vascular artifact. The vertices influenced by large pial veins were first defined on the cortical surface, and then voxels were removed from the entire columns corresponding to these vertices to avoid sampling bias along the cortical depth. Thus, there should be complete data from all cortical depths for the remaining columns. We defined the feedback connectivity from deep layers to deep layers because it represents strong feedback connections according to literature (Markov et al., 2013; Ullman, 1995) and also avoids confounding the feedforward signals from superficial layers.

Markov, N. T., Vezoli, J., Chameau, P., Falchier, A., Quilodran, R., Huissoud, C., Lamy, C., Misery, P., Giroud, P., Ullman, S., Barone, P., Dehay, C., Knoblauch, K., & Kennedy, H. (2014). Anatomy of hierarchy: feedforward and feedback pathways in macaque visual cortex. The Journal of comparative neurology, 522(1), 225–259. https://doi.org/10.1002/cne.23458

Ullman S. (1995). Sequence seeking and counter streams: a computational model for bidirectional information flow in the visual cortex. Cerebral cortex, 5(1), 1–11. https://doi.org/10.1093/cercor/5.1.1

(4) More detail on other subjects in Figure S1. Ten subjects conducted visual fixation and used a bite bar. Imaging data are illustrated in detail from one subject and the remaining subjects are depicted in graphs and in Supplemental Figure S1. Please provide arrowheads in each image to help guide the reader. Some kind of summary or index of modularity would also be helpful.

Thanks for your suggestions. There are arrowheads in each image in our original manuscript and we have revised Figure S1 for better illustration. Additionally, we have added a table summarizing the number of stripes to provide a clearer overview.

(5) How are ROIs in V3ab and V4 defined? V2 ROIs were defined (thin, thick, and pale stripe), but V3ab and V4 averaged across the whole area. Why not use the most activated "domains" from V3ab and V4? How does this influence connectivity analysis?

Thank you for your question. We defined V4 and V3ab on the cortical surface using a retinotopic atlas (Benson 2018), which has been shown to be quite accurate in defining ROIs for the early visual areas. Since all ‘domains’ showed robust BOLD activation to our stimuli, we used voxels from the entire ROI in the depth-dependent analysis. In the functional connectivity analysis, we used the most informative voxels by feature selection, which should already be included in the feature domains.

Minor:English language editing is needed.

Thank you for your feedback. We have carefully revised the manuscript for clarity and readability.

Line 31 "its" should be "their".

Thank you. We have corrected "its" to "their".

Replace 'representative subject' with 'subject'.

We have replaced "representative subject" with "subject" in the manuscript.

Replace 'naturalistic texture' with 'texture'.

Thank you for your suggestion. The textures used in our experiment were generated based on the algorithm by Portilla and Simoncelli (2000), and the term "naturalistic texture" was used to be consistent with literature. The textures used in our study are different from traditional artificial textures, as they contain higher-order statistical dependencies. Following your recommendations, we have replaced ‘naturalistic texture’ with ‘texture’ in some places in the main text to improve readability.

Typo: Line 126, Fig 2B should be 1B.

Thank you. We have corrected "Fig 2B" to "Fig 1B" in Line 128.

Fig. 2A: point out where are texture domains in anterior V2.

The texture-selective activations in anterior V2 (corresponds to peripheral visual field) have been highlighted by arrowheads.

Fig 2B, 3 legend: Round symbols are for each subject?

Yes, the round symbols in Figures 2B represent data for individual participants. We have revised the legend for clarity.

Fig. 3: Disparity and texture values do not look different across depth (except may the V2 texture values).

While the difference in feature selectivity is small across cortical depths, they are highly consistent across participants. We have provided a figure showing the original BOLD responses in the revised manuscript (Figure S8 and Figure S8). Data from individual subjects were also available at Open Science Framework (OSF, https://doi.org/10.17605/OSF.IO/KSXT8 (‘rawBetaValues.mat’ in the data directory)).

Line 57-59 The statement is not strictly accurate. V1 also has color, orientation, and motion representations.

Thank you for your feedback. Our statement was intended to convey that M and P information from the geniculate input are transformed into representations of color, orientation, disparity, and motion in the primary visual cortex. We have clarified this point in the revised manuscript.

In Line 58-60: “In the primary visual cortex (V1), the M and P information from the geniculate input are transformed into higher-level visual representations, such as motion, disparity, color, orientation, etc. (Tootell & Nasr, 2017).”

Fig. 1B V1 interblobs also project to thick stripes (Sincich and Horton).

Thank you for the additional information. We appreciate your input. Our figure is intended as a simplified schematic and does not fully represent all the connections. We have discussed this reference in the revised manuscript.

In Line 136-138: “Only major connections were shown here. There are also other connections, such as V1 interblobs projecting to thick stripes (Federer et al., 2021; Hu & Roe, 2022; Sincich and Horton, 2005).”

Line 207 "suggesting that both local and feedforward connections are involved in processing color information in area V2." Logic? English?

Thank you for pointing this out. The superficial layers are involved in local intracortical processing by lateral connections and also send output to higher order visual areas along the feedforward pathway. Thus, the strongest color selectivity in the superficial depth of V2 supports that color information was processed in local neural circuits in area V2 and transmitted to higher order areas along the feedforward pathway. We have revised the manuscript for clarity.

In Line 241-245: “According to the hierarchical model, the strongest color selectivity in the superficial cortical depth is consistent with the fact that color blobs locate in the superficial layers of V1 (Figure 1B, Felleman & Van Essen, 1991; Hubel & Livingstone, 1987; Nassi & Callaway, 2009). The strongest color selectivity in superficial V2 suggests that both local and feedforward connections are involved in processing color information (Figure 1C).”

Line 254 "Laminar". Please use "cortical depth" or explicitly state that 'laminar' refers to superficial, middle, and deep as defined by cortical depth.

Thank you for your suggestion. We have clarified the term "laminar" in the manuscript as referring to superficial, middle, and deep layers as defined by cortical depth.

In Line 96-99: “To better understand the mesoscale functional organizations and neural circuits of information processing in area V2, the present study investigated laminar (or cortical depth-dependent) and columnar response profiles for color, disparity, and naturalistic texture in human V2 using 7T fMRI at 1-mm isotropic resolution.”

Fig. S5 Please add a unit of isoluminance.

Thank you for your suggestion. Supplementary Figure S10A and S10B illustrate the blue-matched luminance levels in RGB index. In our isoluminance experiment, blue was set as the reference color (RGB [0 0 255]) to measure the red and gray isoluminance.

Line 448-449 To make this rationale clearer, refer to:Wang J, Nasr S, Roe AW, Polimeni JR. 2022. Critical factors in achieving fine‐scale functional MRI: Removing sources of inadvertent spatial smoothing. Human Brain Mapping. 43:3311-3331.

Thank you for your suggestion. We have added this reference to better support the rationale of data analysis.

**Reviewer #2:**
(1) Line 126 should refer to Figure 1B.

Thank you. We have corrected the reference in the revised manuscript as Figure 1B.

(2) Even if only one naturalistic texture session was acquired per participant, it might be interesting to see the within-session repeatability by, e.g., splitting the texture runs into two halves.

Thank you for your suggestion. We performed a split-half correlation analysis for participants who completed 10 runs in the naturalistic texture session. The result from one representative subject was shown in the figure below (for other participants, r = 0.38, 0.38, 0.24, and 0.23, respectively).

**Author response image 2. sa4fig2:** Split-half correlations for the texture-selective activation maps in a representative subject (S01) in V2.

(3) Unfortunately, Figure S2 only shows the stripe ROIs but not V3ab or V4 ROIs. Including another figure that shows all ROIs in more detail would be interesting.

Thank you for your suggestion. We have included a figure showing the ROIs for V4 and V3ab (the black dotted lines in Figure S9).

(4) It would be helpful for the reader to have a more detailed discussion about methodological limitations, including the unspecificity of the GE-BOLD signal (Engel et al., 1997, Cereb Cortex, 7, 181-192; Parkes et al., 2005, MRM, 54, 1465-1472; Fracasso et al., 2021, Prog Neurobiol, 202, 102187) and the used voxel sizes.

Thank you for your suggestion. We have added a more detailed discussion about the methodological limitations, including the unspecificity of the GE-BOLD signal and the voxel sizes used.

In Line 397-408: “Due to the limitations of the T2*w GE-BOLD signal in its sensitivity to large draining veins (Fracasso et al., 2021; Parkes et al., 2005; Uludag & Havlicek, 2021), the original BOLD responses were strongly biased towards the superficial depth in our data (Figure S8). Compared to GE-BOLD, VASO-CBV and SE-BOLD fMRI techniques have higher spatial specificity but much lower sensitivity (Huber et al., 2019). As shown in a recent study (Qian et al., 2024), using differential BOLD responses in a continuous¬¬ stimulus design can significantly enhance the laminar specificity of the feature selectivity measures in our results (Figure 3). Compared to the submillimeter voxels, as used in most laminar fMRI studies, our fMRI resolution at 1-mm isotropic voxel may have a stronger partial volume effect in the cortical depth-dependent analysis. However, consistent with our results, previous studies have also shown that 7T fMRI at 1-mm isotropic resolution can resolve cortical depth-dependent signals in human visual cortex (Roefs et al., 2024; Shao et al., 2021).”

(5) If I understand correctly, different numbers of runs/sessions were acquired for different subjects. It would be good to discuss if this could have impacted the results, e.g., different effect sizes could have biased the manual ROI definition.

Thank you for your suggestion. Although there were differences in the number of runs/sessions acquired for different subjects, there were at least four runs of data for each experiment, which should be enough to examine the within-subject effect. We have discussed this point in the revised manuscript.

In Line 481-484: “Although the number of runs were not equal across participants, there were at least four runs (twenty blocks for each stimulus condition) of data in each experiment, which should be sufficient to investigate within-subject effects.”

(6) It would be good to add the software used for layer definition. Was it Laynii?

We have provided more details in the revised methods.

In Line 523-526: “An equi-volume method was used to calculate the relative cortical depth of each voxel to the white matter and pial surface (0: white matter surface, 1: pial surface, Supplementary Figure S11A), using mripy (https://github.com/herrlich10/mripy).”

(7) It would be interesting to see (at least for one subject) the contrasts of color-selective thin stripes and disparity-selective thick stripes from single sessions to demonstrate the repeatability of measurements.

Thank you for your suggestion. We have shown the test-retest reliability of the response pattern of color-selective thin stripes and disparity-selective thick stripes in a representative subject in Figure S5.

(8) By any chance, do the authors also have resting-state data from the same subjects? It would be interesting to see the connectivity analysis between stripes and V3ab, V4 with resting-state data.

Thank you for your suggestion. Unfortunately, we do not have resting-state data from the same subjects at this time. We agree with you that layer-specific connectivity analysis with resting-state data is very interesting and worth investigating in future studies.

**Reviewer #3:**
(1) For investigating information flow across areas, the authors rely on layer-specific informational connectivity analyses, which is an exciting approach. Covariation in decoding accuracy for a specific dependent variable between the superficial layers of a lower area and the middle layer of a higher area is taken as evidence for feedforward connectivity, whereas FB was defined as the connection between the two deep layers. Yet this method is not assumption-free. For example, the canonical idea (Figure 1C) of FF terminals exclusively arriving in layer 4 and FB terminals exclusively terminating in supra-or infragranular layers is not entirely correct. This is not even the case for area V1 - see for example Kathy Rockland's exquisite tractography studies, showing that even single axons with branches terminating in different layers. Also, feedback signals not only arrive in the deep layers of a lower area. Although these informational connectivity analyses can be suggestive of information flow, this reviewer doubts it can be considered as conclusive evidence. Therefore, the authors should drastically tone down their language in this respect, throughout the text. They present suggestive, not conclusive evidence. To obtain truly conclusive evidence, one likely has to perform laminar electrophysiological recordings simultaneously across multiple areas and infer the directionality of information flow using, for example, granger causality.

Thank you for pointing out this important issue. In our response to a previous question (Reviewer #1, the 2nd comment), we have discussed other possible connections in addition to the canonical feedforward and feedback pathways. In the revised manuscript, the conclusion has been toned down to properly reflect our findings. However, we would also like to emphasize that our conclusion about laminar circuits was supported by converging lines of evidence. For example, in addition to the depth-dependent connectivity results, the role of feedback circuit in processing texture information was also supported by greater selectivity in V4 than V2, and the strongest deep layer selectivity in V2 (Figure 3C).

(2) In the same realm, how reproducible are the information connectivity results? In the first part of the study, the authors performed a split-half analyses. This should be also done for Figure 4.

Thank you for your suggestion. We have performed a split-half analysis for the informational connectivity results. As shown in Author response image 3, the results for the color experiment were robust and reproducible, while the disparity and texture connectivity results were less consistent between the two halves. The results from the second half (Author response image 3, below) are more consistent with the original findings (Figure 4). Overall, the pattern of results were qualitatively similar between the two halves. The inconsistency may be due to the fact that some participants had only four runs of data, which could make the split-half analysis less reliable.

**Author response image 3. sa4fig3:** Split-half analysis of informational connectivity.

(3) Most of the other layer-specific claims (not the ones about the flow of information) are based on indices. It is unclear which ROIs contributed to these indices. Was it the entire extent of V1, V2, ...? Or only the visually-driven voxels within these areas? How exactly were the voxels selected? For V2, it would make sense to calculate the selectivity indices independently for the disparity and color-selective (putative) thick and (putative) thin stripe compartments, respectively. Adding voxels of non-selective compartments (e.g. putative thick stripe voxels for calculating the color-index; or adding putative thin-strip voxels for calculating the disparity index), will only add noise.

In the revised manuscript, we have clarified that we selected the entire ROI in the depth-dependent analysis. Since our study does not have an independent functional localizer, using the entire ROI avoids the problem of double dipping. The processing of visual features is not confined solely to specific stripes. We have also provided a more comprehensive explanation of this issue in the discussion section.

In Line 541-544: “For the cortical depth-dependent analyses in Figure 3, we used all voxels in the retinotopic ROI. Pooling all voxels in the ROI avoids the problem of double-dipping and also increases the signal-to-noise ratio of ROI-averaged BOLD responses.”

(4) It is apparent from Figure 3, that the indices are largely (though not exclusively) driven by 2 subjects. Therefore, this reviewer wishes to see the raw data in addition to a table for calculating the color, disparity, and texture selectivity indices -along with the number of voxels that contributed to it.

Thank you for your suggestion. We have provided a figure showing the original BOLD responses (Figure S8 and Figure S8). Data from individual subjects were also available at Open Science Framework (OSF, https://doi.org/10.17605/OSF.IO/KSXT8 (‘rawBetaValues.mat’ in the data directory)).

Minor:(1) I typically find inferences about 'layer fMRI' vastly overstated. We all know that fMRI does not (yet) provide laminar-specific resolution, i.e., whereby meaningful differences in fMRI signals can be extracted from all 6 individual layers of neocortex, without partial volume effects, or without taking into account pre-and postsynaptic contributions of neurons to the fMRI signal (the cell bodies may very well lay in different layers than the dendritic trees etc.), or without taking into account the vascular anatomy, etc. The authors should use the term cortical depth-dependent fMRI throughout the text -as they do in the abstract and intro.

Thank you for pointing out this important issue. We have now defined the meaning of layer or laminar as “cortical depth-dependent” in the introduction, to be consistent with the terminology in most published papers on this topic.

(2) 1st sentence abstract: I disagree with this statement. The parallel streams in intermediate-level areas are probably equally well studied as the geniculostriate pathway -already starting with the seminal work of Hubel, Livingstone, and more recently by Angelucci and co-workers who looked in detail at the anatomical and functional interactions across sub-compartments of V1 and V2.

Thank you for your feedback. In the revised manuscript, we have removed the term "much" from the first sentence of the abstract. Although there have been seminal studies of V2 sub-compartments in monkeys, only a few fMRI studies investigated this issue in humans.

(3) The authors show inter-session correlations for color and disparity. This reviewer would like to see test-retest images since the explained variance is not terribly good. Also, show the correlation values for the inter-session texture beta values.

Thank you for your suggestion. We have performed the test-retest reliability analysis of texture-selective patterns in the response to a previous question (Reviewer #2, the 2nd comment, Author response image 2).

(4) The stripe definitions are threshold dependent. Please clarify whether the reported results are threshold-independent.

Thank you for your question. To address your concern, we defined the stripe ROIs using different thresholds, and the results remained consistent. Specifically, we ranked the voxels in manually defined stripe ROIs by the color-disparity response. We then defined the lowest 10% as the thick stripe voxels, the highest 10% as thin stripe voxels, and the middle 10% as pale stripe voxels. Additionally, we adjusted the thresholds to 20% and 30% to define the three stripes (with 30% being the least strict threshold). Feature selectivities at different thresholds were shown in Figure S6 (from left to right: 10%, 20%, 30%). Notably, in all threshold conditions, there was no significant difference in texture selectivity across different stripes.

(5) How were the visual areas defined?

In the revised manuscript, we have provided a detailed description about methods.

In Line 531-535: “ROIs were defined on the inflated cortical surface. Surface ROIs for V1, V2, V3ab, and V4 were defined based on the polar angle atlas from the 7T retinotopic dataset of Human Connectome Project (Benson et al., 2014, 2018). Moreover, the boundary of V2 was edited manually based on columnar patterns. All ROIs were constrained to regions where mean activation across all stimulus conditions exceeded 0.”

(6) "According to the hierarchical model in Figure 1B and 1C, the strongest color selectivity in the superficial cortical depth is consistent with the fact that color blobs mainly locate in the superficial layers of V1, suggesting that both local and feedforward connections are involved in processing color information in area V2." But color-selective activation within V2 could be also consistent with feedback from other areas (some of which were not covered in the present experiments) -the more since most parts of the brain were not covered (i.e. a slab of 4 cm was covered)?

Thank you for reminding us about this issue. We have discussed the possibility of feedback influence in explanation of the superficial bias of color selectivity in area V2.